# Finite Differences for Recovering the Plate Profile in Electrostatic MEMS with Fringing Field

**Mario Versaci** [1,*] , **Luisa Fattorusso** [2] , **Alessandra Jannelli** [3] **and Paolo Di Barba** [4]

1 DICEAM Department, "Mediterranea" University, I-89122 Reggio Calabria, Italy
2 DIIES Department, "Mediterranea" University, I-89122 Reggio Calabria, Italy
3 MIFT Department, University of Messina, I-98166 Messina, Italy
4 Dipartimento di Ingegneria Industriale e dell'Informazione, University of Pavia, Via A. Ferrata 5, I-27100 Pavia, Italy
* Correspondence: mario.versaci@unirc.it; Tel.: +39-09651692273

**Abstract:** Global existence and uniqueness conditions for a dimensionless fourth-order integro-differential model for an electrostatic-elastic MEMS device with parallel plates and fringing field contribution were recently achieved by the Authors. Moving from this work, once the dielectric profile of the deformable plate according with experimental setups has been assigned, some technical conditions of applicability for the intended use of the device as well as the mechanical tension of the deformable plate are presented and discussed. Then, highlighting the link between the fringing field and the electrostatic force, finite differences were exploited for recovering the deformable plate profile according both global existence and uniqueness conditions. Moreover, the influence of the electro-mechanical properties of the deformable plate on both the numerical approach and on the intended uses of the device is discussed, comparing the results with experimental setups regarding pull-in voltage and electrostatic pressure.

**Keywords:** 3D electrostatic-elastic MEMS; Pelesko–Driskoll's theory for fringing field modeling; finite difference approaches; profile recovering; ghost solutions

## 1. Introduction

In the last decade, a high synergy between the skills of scientific research and the world of industry has emerged for the development of technologically transferable physical-mathematical models capable of formalizing the increasingly performing behaviors of micro-electro-mechanical systems (MEMS) [1–3]. Today, these devices are considered "intelligent" because they combine electrical, electronic, mechanical, optical and other behaviors, managing highly complex industrial processes [4–9]. Among these, electrostatic MEMS with parallel metal plates are widely used devices in the industry, as they are easy to construct and exhibit high versatility [1,10,11]. Today, theoretical modeling of these devices is so advanced as to evaluate any correspondences between mechanical stresses of the deformable elements and the intended use of the device without neglecting the possibility of carrying out functionality tests in operation, which would normally require the destruction of the device itself [12,13]. Whatever the intended use of the device, it is necessary that there are no electrostatic discharge phenomena inside it, caused by contact between deformable and fixed elements, which would damage the device itself [14–16]. Therefore, it appears necessary to reduce, as much as possible, the physical causes, such as the fringing field, to produce an excessive approach between deformable and fixed elements [17–19]. The fringing field, which strongly depends on the length/width ratio of the device, produces important effects on the bending of the lines of force of the electric field, **E**, inside it, manifesting this influence near the edges; however, in the center, this effect is almost nil [20,21]. Among these models is an important dimensionless integro-differential model of the fourth order, studied with a high level of attention to detail in [22],

which, being endowed with global existence results, opened interesting scenarios regarding future developments. However, due to its formulation, the dimensionless model studied in [22] is not very suitable for technology transfer, since some of its analytical elements do not correctly model the behavior of some elements present in the device. This impasse was overcome in [17] where, starting from [22], a new formulation of the dimensionless model was presented and studied, more in keeping with the industrial reality of the MEMS devices produced, which also considers the effects due at the fringing field. This dimensionless model takes the form:

$$
\begin{cases}
\Delta^2 u(x) = \left( \beta \int_\Omega |\nabla u(x)|^2 \mathrm{dx} + \gamma \right) \Delta u(x) + \\
+ \dfrac{\lambda f(x)}{(1-u(x))^2 \left( 1 + \chi \int_\Omega \frac{\mathrm{dx}}{(1-u(x))} \right)^2} + \lambda \delta |\nabla u(x)|^2 \\
u(x) = 0, \quad \nabla u(x) = 0 \quad x \in \partial\Omega, \\
0 < u(x) < 1 \quad x \in \Omega \subset \mathbb{R}^N, \quad N < 4.
\end{cases}
\tag{1}
$$

where

- $\Omega$ represents the smooth bounded domain (MEMS device);
- $u : \Omega \to \mathbb{R}$ defines the profile of the deforming plate;
- $f : \Omega \to \mathbb{R}^+$ is a bounded function which considers the dielectric properties of the material constituting the deformable plate;
- $\lambda$ is a positive dimensionless parameter depending on the applied voltage, $V$, defined as in (3) which represents the ratio of a reference electrostatic force to a reference elastic force;
- $\beta$ is a positive dimensionless parameter which takes into account the stiffness of the deformable plate;
- $\gamma$ is a positive dimensionless parameter which takes into account the stretching effect in the deformable plate;
- $\chi$, defined as in (8), is a positive dimensionless parameter which takes into account the non-local dependence of $V$ on the solution due to a possible non-uniform electric charge distribution;
- $\delta$ is a positive dimensionless parameter (usually constant) that weighs the effects due to the fringing field.
- $\lambda \delta |\nabla u(x)|^2$ takes into account, according to the Pelesko–Driscoll theory, the effects due to the fringing field [17,23].

In the past, simplified physical-mathematical models, containing terms due to the fringing field but poorly adhering to industrial realities, have been proposed, studied, and validated [23,24].

Model (1) appears interesting for many reasons. On one hand, $\lambda \delta |\nabla u(x)|^2$ allows easy software/hardware implementations, while on the other hand, it is voltage controllable (presence of $\lambda$ that, as specified in (3), depends on $V$), even if the dependence of $\delta$ on the applied external voltage has not yet been highlighted (satisfactory limitations of $\delta$ have been obtained for membrane MEMS devices [17,23,24]). However, even if the stability of (1) has not yet been studied (for which any unstable positions of the deformable plate would risk causing unwanted electrostatic discharges), the device is controlled in voltage (presence of $\lambda$). In such cases, appropriate destinations of use (such as biomedical ones, at reduced voltage) reduce the risk of electrostatic discharges. We also observe that, in (1), there are terms due to stiffness and self-stretching (identifiable by the parameters $\beta$ and $\gamma$) of the deformable element to take into account the mechanical phenomena of fatigue due to the continuous rising-lowering of the deformable element. Furthermore, with the deformation of the plate, significant variations of electrostatic capacitance occur inside the MEMS, making it necessary to express its dependence on the geometric parameters of the device and on an auxiliary electrostatic capacitance ($C_f$) capable of opposing sudden

voltage variations applied (presence of $\chi$ in (1)) [1,17]. Finally, $f(x)$ in (1) (present in the "capacitive" term of the model) models the dielectric properties of the plates.

Equation (1) does not allow to obtain $u(x)$ explicitly, thus in [17], important conditions of global existence and uniqueness for the solution have been obtained by opening possible scenarios for the numerical reconstruction of $u(x)$ profiles without representing ghost solutions (i.e., approximate solutions not satisfying the conditions of existence and uniqueness for (1)). Therefore, the main aims of this paper can be summarized in the following items:

- Exploiting the conditions of global existence and uniqueness for (1), fundamental results are discussed highlighting that, starting from the reference energy state, the solution is unique ensuring that the profile of the deformable plate is uniquely determined without highlighting anomalies in the curvature of the deformable plate (especially in the start-up phase). This item is very important because, compared to what is already known in the literature, the existence and uniqueness of the solution for (1) is studied starting from a reference energy state, thus linking the important properties of existence and uniqueness with the minimum energy level necessary to move the deformable plate.
- After selecting the dielectric profile, $f(x)$ (in order to satisfy binding physical requirements allowing comparison with experimental setups known in the literature), we discuss important industrial implications in terms of limitations both for the applied voltage, $V$, and for the mechanical tension of the deformable plate at rest, $T$, providing both a graphical differentiation of areas allowed in the plane $(V, T)$ and reference energy states. So, unlike what is already known in the literature (where in the dimensionless mathematical models $f(x)$ is set equal to the unit), in this paper we provide a useful criterion for selecting the intended use of the device, based on $(V, T)$, starting at $f(x)$.
- We study how the fringing field affects the electrostatic force in the device, obtaining an increase for it depending on both $V$ (which affects the intended use of the device) and $T$ (which influences the choice of material of the deformable plate). In the literature, there are no increases for either $V$ or $T$, thus failing to provide maximum admissible values for both. This item of the present work fills this gap.
- Following a simplification in the model that does not affect the goodness of the results obtained, the numerical recovering of the deformable plate profile was obtained by the finite difference method (implemented in MatLab® R2019a running on an Intel Core 2 CPU at 1.45 GHz) "gold standard" for model as (1), under different operating conditions, giving results compatible with the conditions of global existence and uniqueness of the solution (thus not representing ghost solutions).
- We also provide an effective criterion for choosing the intended use of the device (strongly linked to $V$) starting from the choice of the material constituting the deformable plate (strongly dependent on $T$) and vice versa very useful for any industrial applications (unlike the scientific works known in the literature where such a criterion has never been elaborated).
- Furthermore, how the electro-mechanical properties of the deformable plate affect the numerical profile recovering is studied and discussed also selecting the most important intended uses of the studied device.
- Finally, important comparative results with experimental setups concerning pull-in voltage and electrostatic pressure are presented.

The reminder of the paper is organized as follows: Beginning with the description of the device and analyzing both behavior and analytical models (Section 2), the most important results of global existence and uniqueness for the solution for (1) are summarized in Section 3. Next, after selecting the dielectric profile of the deformable plate as specified in Section 4 (according to several industrial applications), Section 5, starting the above-mentioned inequality governing the global existence and uniqueness for (1), discussions and important limitations for both the intended use of the device and the mechanical properties of the deformable plate are deduced, respectively. Once Section 6 formalizes the

link between the fringing field and electrostatic force in the device, a numerical approach is proposed and applied in Section 7 to recover the profile of the deformable plate, the results of which are presented and discussed in Section 8. Section 9 discusses an interesting limitation for the mechanical tension of the deformable plate, while Sections 10 and 11, respectively, discuss some details concerning the possible influence of the properties of the deformable plate on the numerical procedure and the possible intended uses of the device. Finally, Section 12 presents the results concerning the pull-in voltage and the electrostatic pressure. Further conclusions and perspectives on these issues complete this work.

## 2. The Behavior of the MEMS Device: The Analytical Model with Fringing Field

The electrostatic MEMS studied in this paper is an elastic system consisting of two parallel plates (of the same material and equal thickness) of which the upper one (non-deformable) is at $V > 0$ electric potential, while the lower plate (bound to the edges and at electric potential reference, $V = 0$) deforms towards the top plate without touching it (to avoid unwanted electrostatic discharges) (for further details, ref. [17]). Figure 1 displays a $3D$ schematic of the device (the deformable plate is in its rest position).

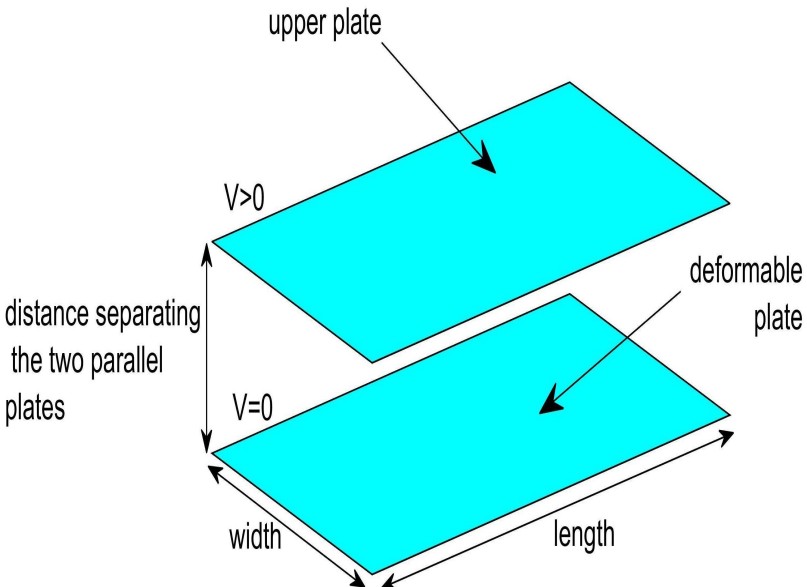

**Figure 1.** Schematic representation of the electrostatic MEMS under study.

As already detailed in [17], if $V$ is the drop voltage, the load function $F(x)$ (which establishes how the deformable plate is stressed when $V$ is applied, determining $\mathbf{E}$ which locally activates an electrostatic pressure deforming the plate) can be written as [17]

$$F(x) = \frac{\lambda f(x)}{(1 - u(x))^2},$$ (2)

with

$$\lambda = \frac{\epsilon_0 V^2 L^2}{2d^3 T},$$ (3)

where $\epsilon_0$ is the permittivity of the free space, $d$ is distance between the plates, and $L$ is the length of the MEMS. (2), according the Pelesko's procedure [1,25], is necessary to formulate

$$K_1(x)\Delta^2 u(x) = K_2(x)\Delta u(x) + F(x) = K_2(x)\Delta u(x) + \frac{\lambda f(x)}{(1 - u(x))^2}.$$ (4)

where $K_1(x)$ and $K_2(x)$ are specific weight functions as below defined. It should be noted that (2) cannot electrically control the MEMS because the applied V must be controlled in order to avoid sudden deformation of the deformable plate. Then, a capacitive control de-

vice (such as the one hypothesized in [17]) can be used to avoid this drawback. Particularly, considering the series of a source voltage, $V_s$, and a capacitor, $C_f$, it follows that

$$V = \frac{V_s}{1 + \frac{C}{C_f}} \tag{5}$$

in which [17]

$$C \approx \frac{\epsilon L^2}{d} \int_\Omega \frac{dx}{1 - u(x)} \tag{6}$$

achieving

$$V = \frac{V_s}{1 + \frac{\frac{\epsilon L^2}{d} \int_\Omega \frac{dx}{1-u(x)}}{C_f}}. \tag{7}$$

Therefore, setting

$$\chi = \frac{\epsilon_0 L^2}{C_f d}, \tag{8}$$

from (7), we can write

$$\underbrace{\left(\frac{V}{V_s}\right)^2}_{\text{control action}} = \frac{1}{\left(1 + \chi \int_\Omega \frac{dx}{(1-u(x))}\right)^2}. \tag{9}$$

Finally, (2) becomes

$$F(x) = \frac{\lambda f(x)}{(1 - u(x))^2} \left(\frac{V}{V_s}\right)^2 = \frac{\lambda f(x)}{(1 - u(x))^2 \left(1 + \chi \int_\Omega \frac{dx}{(1-u(x))}\right)^2} \tag{10}$$

so that (4) assumes the final form

$$K_1(x)\Delta^2 u(x) = K_2(x)\Delta u(x) + \frac{\lambda f(x)}{(1 - u(x))^2 \left(1 + \chi \int_\Omega \frac{dx}{(1-u(x))}\right)^2}. \tag{11}$$

We also observe that $\chi$ mainly depends on the capacity [1] and, furthermore, $\chi \in [0, 1)$ [17], thus avoiding a dangerous bifurcation phenomena which can make the device highly unstable (particularly if $\chi \to 1^-$ the rupture of the device can takes place [17]). Finally, $\Delta u(x)$ being an operator indicative of the curvature assumed by the plate during deformation [26], it can be proved that $K_1(x)\Delta^2 u(x)$, in elastic regimes, takes the form [17]:

$$K_1(x)\Delta^2 u(x) = \underbrace{\left(\beta \int_\Omega |\nabla u(x)|^2 dx + \gamma\right)}_{K_2(x)} \Delta u(x) + \tag{12}$$

$$+ \frac{\lambda f(x)}{(1 - u(x))^2 \left(1 + \chi \int_\Omega \frac{dx}{(1-u(x))}\right)^2},$$

and being $K_1(x) = \frac{D}{L^2 T} \approx 1$ [1,17,26] ($D$, flexural rigidity of the material constituting the deformable plate) achieves the following model:

$$\begin{cases} \Delta^2 u(x) = \left(\beta \int_\Omega |\nabla u(x)|^2 dx + \gamma\right) \Delta u(x) + \frac{\lambda f(x)}{(1-u(x))^2 \left(1 + \chi \int_\Omega \frac{dx}{(1-u(x))}\right)^2} \\ u(x) = 0, \quad \nabla u(x) = 0 \quad x \in \partial\Omega, \\ 0 < u(x) < 1 \quad x \in \Omega \subset \mathbb{R}^3 \end{cases} \tag{13}$$

studied in [1,25], highlighting important results regarding the bifurcation. Finally, to take into account the effects due to the fringing field (physical phenomenon, heavily dependent on the ratio $L/d$, that manifests itself with the curvature of the **E** lines of force, more evident at the edges of the device and negligible in its center), we use the Pelesko–Driscoll formulation [23] that quantifies these effects through the addend $\lambda\delta|\nabla u(x)|^2$ (for details, [17,22,23]), achieving model (1). Table 1 lists all symbols present in (1).

**Table 1.** List of Dimensionless Parameters in Model (13).

| Symbol | Meaning |
| --- | --- |
| $f(x)$ | dielectric profile of the deformable plate |
| $\beta$ | parameter that weighs the deformation energy of the deformable plate |
| $\lambda$ | ratio of a reference electrostatic force to a reference elastic force |
| $\gamma$ | stretching parameter |

We observe that, under the effect of the fringing field, the electrostatic capacity of MEMS varies considerably and can also be formulated through empirical approaches [1,18,25]. Recently, many MEMS models with fringing field have been studied according to the Pelesko–Driscoll theory [24] (which, however, propose studies on highly simplified models). Obviously, even if (1) models a detailed behavior of the MEMS under study, it does not consider further non-linearities (local mechanical stresses, bifurcations), and thus introduces a degree of uncertainty. However, to take into account local mechanical stresses would mean to introduce in the model a terms depending, for example, on the Piola-Kirchhoff tensor which, usually, exhibits discontinuities along the surface of separation between different zones of the deformable plate. This additional term would reduce the amplitude of the deformation of the membrane profile by approximately 5%. Therefore, not considering the contribution due to this term allows us to affirm that the numerical recovering we will obtain will be overestimated but for the sake of safety since we are sure that the real deformation of the deformable plate is below that reconstructed numerically (certainty that the deformable plate does not touch the upper wall of the device). Furthermore, considering $\lambda < \lambda^*$ ($\lambda^*$, pull-in voltage) assures us that bifurcation phenomena cannot take place.

**Remark 1.** *It is worth noting that model (1) represents of a generalization of the model extensively studied in [27]. In fact, when $\beta = \gamma = \chi = 0$ (i.e., in the presence of a deformable membrane), it is easy to achieve*

$$\Delta^2 u(x) = \frac{\lambda f(x)}{(1-u(x))^2} + \lambda\delta|\nabla u(x)|^2 \qquad (14)$$

*which represents the model studied in [27].*

**Remark 2.** *From (3), it is easy to deduce that by selecting the type of device (that is, by choosing T), its intended use is determined ($\lambda$ is proportional to $V^2$). The converse is also valid: once the use of MEMS has been identified, the material constituting the deformable plate can be selected (in other words, the intended use of the device sets an important limitation for $\lambda$). Therefore, as already verified in the past for other devices [28], the link between the intended use of the device and the mechanical properties of the deformable plate is confirmed.*

**Remark 3.** *Equation (2), as already mentioned, represents the type of load on the deformable plate applying V externally (V determines **E** between the two plates by generating the electrostatic force which, per surface unit, is transformed into electrostatic pressure acting on the deformable plate). Then, the need for (2) to depend on V is confirmed.*

**Remark 4.** *We observe that the dielectric properties of the material constituting the deformable plate are decisive for the operation of the device when the deformation is the maximum allowed [1,25]. In fact, indicating with d\* the minimum distance allowed between the two plates, it follows that*

$u(x) < 1 - d^*$ *(this is due to the fact that, physically, the deformable plate must not touch the upper plate and, mathematically, in* (1) $u(x) < 1$.*). If* $u(x) = 1 - d^*$, *from the equation of model* (1), *it follows that* $f(x) = 0$, *so that* (2) *is canceled. Industrially, this is highly important, because* $f(x)$ *contributes to determine the electrostatic load in the device. Therefore,* (1) *represents a good formulation for the device, since it takes into account* $f(x)$ *(as already experimentally tested for similar devices [29]).*

**Remark 5.** *Equation* (1) *models electrostatic MEMS for industrial applications that require small displacements of the deformable plate* ($u(x) \approx 10^{-6}$). *In fact, if* $A \gg d^2$ *(d the distance between the two plates at rest, A the surface of the plates) the effects due to the fringing field would be negligible (i.e.,* $\lambda\delta|\nabla u(x)|^2 = 0$). *However, from* $A \gg d^2 > u^2(x)$ *it would follow* $u(x) \ll \sqrt{A} \approx 10^{-6}$ *which does not fit well with the hypothesis of small displacements. This confirms that* (1) *is a good model for industrial developments.*

**Remark 6.** *Unlike other analytical models [29,30], Equation* (1) *was built on very simple device geometry; this was necessary to carry out the analytical study [17] obtaining results which, even if they have not yet achieved an experimental confirmation, have provided a significant theoretical contribution.*

### 3. An Important Result of Global Existence and Uniqueness

The main global existence and uniqueness result for (1) was presented in [17] (However, Equation (1) does not allow to achieve explicit solutions). For simplicity of reading, we report the fundamental Theorem.

**Theorem 1.** *Let us consider a smooth bounded domain* $\Omega \subset \mathbb{R}^N$, *with* $N < 4$ *on which to consider the problem* (1). *Moreover, let us consider* $f(x) \in L^\infty(\Omega)$ *and* $\alpha$, $\beta$, $\gamma$, $\chi > 0$. *Then, there exists,* $\lambda^*$ $\ni' \forall \lambda \in (0, \lambda^*)$, *problem* (1) *has a solution* $u \in H^4(\Omega)$ *with the diameter of* $\Omega$, $d_\Omega$, *sufficiently small* ($d_\Omega \ll 1$) *and* $\delta \in (0, +\infty)$.

We note that $\Omega$ (deformable plate), industrially, has dimensions of the order of $10^{-6}$. Therefore, the diameter of the domain can be considered $\ll 1$ (as predicted by Theorem 1). Furthermore, the constraint $N < 4$, even if more stringent than $N < 8$ [31], does not affect the industrial validity of the result (which requires $3D$ formulations).

The fact that the result requires $f(x) \in L^\infty(\Omega)$ is equivalent to stating that $f(x)$ is measurable, i.e.,

$$||f||_\infty = \inf\{S \geq 0 \ : \ |f(x)| \leq S \ a.e.\}. \tag{15}$$

Then, for each point of the deformable plate, $|f(x)|$ must be bounded. So, (15) makes sense because $f(x)$ represents the dielectric profile of the deformable plate which must be measurable (and in any case positive and bounded). Furthermore, Theorem 1 dictates that $\exists \lambda^* \ni' \forall \lambda \in (0, \lambda^*) : \lambda < \lambda^*$ because $\lambda^*$, is pull-in voltage (i.e., $\lambda$ such that there are no solutions for $\lambda \geq \lambda^*$). Finally, the continuity of higher order curvatures ($u \in H^4(\Omega)$) is required by imposing that the deformable element, during its movement, must not undergo substantial deformations such as to make the device unusable (especially when fatigue phenomena caused by prolonged and continuous exploitation take place [26]). Obviously, this condition is more restrictive with respect to the conditions required when membrane MEMS are considered. It is worth nothing that, even though $\lambda$ is controllable by $V$, $\delta$ is not; this dependence is expressed by the product $\lambda\delta$ in (1). In the future, this would require an incisive effort on the possible generalization of the Pelesko–Driskoll theory.

The proof of Theorem 1, exploiting a well-known topological result presented in [32], is divided into five points preliminarily defining the following two suitable sets:

$$X = \mathbb{R}^+ \times \mathbb{R}^+ \times \mathbb{R}^+ \times \{f \in L^\infty(\Omega) \ : \ |x \ : \ f(x) > 0| \neq 0\} \tag{16}$$

and

$$Y = \left\{ u \in H^4(\Omega) \cap H_0^1(\Omega) : \quad 0 < u < 1, \quad \int_\Omega \frac{dx}{(1-u)^{16}} < M, \right. \tag{17}$$

$$\left. \int_\Omega |\nabla u|^4 dx < M_2 \quad \text{and} \quad \int_\Omega |\Delta u|^2 dx < M_1, \quad M, M_1, M_2 > 0 \right\}.$$

where $M$, $M_1$ and $M_2$ are suitable positive constants. Moreover, if $Z = L^2(\Omega)$, $B = \Delta^2$ and $x_0 = (0, 0, 0, f_0) \in X$ with $y_0 \in Y$ such that $u_0 = y_0(0, 0, 0, f_0) = y_0(x_0)$ where $u_0$ is the solution of

$$\begin{cases} \Delta u_0 = \frac{\lambda f_0}{(1-u_0)^2} + \lambda \delta |\nabla u_0|^2 & \text{on } \Omega \\ 0 < u_0 < 1 \\ u_0 = 0 & \text{on } \partial\Omega, \end{cases} \tag{18}$$

the following important inequalities governing both global existence and uniqueness for (1) have been achieved:

$$\int_\Omega |F(x, y_0) - F(x_0, y_0)|^2 dx \leq \tag{19}$$

$$\leq 4\beta^2 \left( \int_\Omega |\nabla y_0|^2 dx \right)^2 \int_\Omega |\Delta y_0|^2 dx + 4\gamma^2 \int_\Omega |\Delta y_0|^2 dx +$$

$$+ 4\lambda \|f - f_0\|_{\infty,\Omega}^2 \int_\Omega \frac{1}{(1-y_0)^2} dx + 4|\Omega| \chi^2 \|f_0\|_{\infty,\Omega}^2 \lambda \int_\Omega \frac{1}{(1-y_0)} dx,$$

$$\int_\Omega \left| B(y_1) - B(y_2) - (F((\beta, \gamma, \chi, f, y_1) - F(\beta, \gamma, \chi, f, y_2)) \right|^2 dx \leq \tag{20}$$

$$\leq C(\beta, \lambda, \gamma, \delta, \chi, M, M_1, M_2, f) d_\Omega^{\frac{N}{2}} \int_\Omega \left| B(y_1) - B(y_2) \right|^2 dx,$$

$$\int_\Omega |G(\beta, \gamma, y_1) - G(\beta, \gamma, y_2)|^2 dx \leq \tag{21}$$

$$\leq 2 \left( \beta C d_\Omega^2 M_1 + \gamma \right)^2 C d_\Omega^4 \left( \int_\Omega |\Delta^2(y_1 - y_2)|^2 dx \right) +$$

$$+ 8 C d_\Omega^6 M_1^2 \beta^2 \int_\Omega |\Delta^2(y_1 - y_2)|^2 dx$$

where

$$\begin{cases} G(\beta, \gamma, u(x)) = \left( \beta \int_\Omega |\nabla u(x)|^2 dx + \gamma \right) \Delta u(x) \\ g(\chi, u(x)) = \left( 1 + \chi \int_\Omega \frac{dx}{(1-u(x))} \right)^2 \\ F(\beta, \gamma, \chi, f, y(x), \delta) = \\ = \Delta^2 y(x) - G(\beta, \gamma, y(x)) - \frac{\lambda f(x)}{(1-y(x))^2 g(\chi, y(x))} - \lambda \delta |\nabla y(x)|^2. \end{cases} \tag{22}$$

and $x = (\beta, \gamma, \chi, f)$ with $y = u(x)$.

## 4. Analytical Modeling of $f(x)$

As proved in [33], regardless of how the dielectric constant profile is chose, the pull-in instability cannot be avoided [1,25,29]. Thus, it seems legitimate to ask whether a physically valid formulation of $f(x)$ can influence the solutions. It is worth nothing that in [17] it has been proved that the smooth solution $u(x)$ are symmetric, concave and, moreover, $u(x) < 1$. From the heuristic point of view, the most unstable area of the device is its center,

where the influence of **E** is very strong and the influence in the supporting boundaries appears rather weak [33]. This is due to the fact that $|\mathbf{E}|$ is proportional to

$$\frac{\lambda}{(1 - u(x))^2} \tag{23}$$

so that at the center of the device, by virtue of the symmetry and concavity of the solutions, $1 - (x)$ assumes a minimum value (i.e., maximum value of $|\mathbf{E}|$). Therefore, it seems reasonable to try to reduce $|\mathbf{E}|$ in the center of the device as such possible by allowing it to be stronger near the boundaries (more stable parts of the device). So, by choosing

$$f(x) = |x|^\alpha, \quad \alpha \geq 0 \tag{24}$$

it follows that

$$|\mathbf{E}| \propto \frac{\lambda f(x)}{(1 - u(x))^2} = \frac{\lambda |x|^\alpha}{(1 - u(x))^2} \tag{25}$$

which represents the load function (see (2)). However, we speculate whether specific formulations of $f(x)$ can affect the numerical solutions (and any multiplicity) without neglecting the pull-in voltage and the stable operating range of the device. In this work, $f(x)$ is chosen to satisfy a symmetric power law such as

$$f(x) = |x|^\alpha, \quad \alpha \geq 0. \tag{26}$$

In addition, a large number of experimental and industrial applications are based on (26) [1,33–36]. Finally, for $\alpha = 0$, $f(x) = 1$.

## 5. On the Global Existence of the Solution and Industrial Implications
### 5.1. On the Reference Energy State

Let us first observe that the satisfaction of (18) by $u_0$ means that, in $(x_0, u_0)$, the deformable plate behaves like a membrane. In fact, (18) is a typical semi-elliptical nonlinear model describing an electrostatic membrane MEMS device. This is confirmed by the fact that, by definition,

$$(x, y) = ((\beta, \gamma, \chi, f), u = y(x)) \tag{27}$$

from which

$$(x_0, y_0) = ((\beta = 0, \gamma = 0, \chi = 0, f), u = y(x)). \tag{28}$$

Therefore, $\beta = 0$ means that the energy accumulated by the deformable plate is zero (reference energy state). Moreover, $\gamma = 0$ implies that the effects due to stretching are nil, while $\chi = 0$ imposes that $C_f \to \infty$ with immediate consequence that $V = V_s$ (i.e., the drop-voltage equivalent to the external voltage applied). This is in accordance with the experimental and industrial experience according to which the deformable plate, in the start-up phase, is similar to a membrane [1,25,26].

Furthermore, in [17], it has been shown that $B(Y)$ is a neighborhood of $z_0 = B(y_0)$. Therefore, there exists a ball $S(z_0, r) \subset B(Y)$, with radius $r$, and a neighborhood of $x_0$, $V(x_0)$, such that

$$\begin{cases} F(x, y(x)) = 0 & \forall x \in V(x_0) \\ y(x_0) = y_0 \end{cases} \tag{29}$$

has a single solution

$$y : V(x_0) \to B^{-1}(S(z_0, r)). \tag{30}$$

Then, there exists a neighborhood of $x_0 = (\beta = 0, \gamma = 0, \chi = 0, f_0)$ such that

$$\Delta^2 u(x) = \frac{\lambda f(x)}{(1 - u(x))^2} + \lambda \delta |\nabla u(x)|^2 = 0 \tag{31}$$

and, starting from (31), the solution is unique in the neighborhood of $x_0$. This highlights the fact that, starting from the reference energy state, it is guaranteed that the solution is unique (due also to the fact that the deformable plate behaves like a membrane [1,25,26]).

*5.2. On the Continuity of $x \to F(x, y_0)$*

Considering the elastic deformation regime of the plate valid, the elastic force will be proportional to the increase of the surface, so that the stretching energy is writable as

$$E_s = \left( \beta \int_\Omega |\nabla y_0|^2 \mathrm{d}x \right)^2 + \gamma^2 \tag{32}$$

so that (19) (deriving, according to [32], from the study of the continuity of the mapping $x \to F(x, y_0)$) becomes

$$\int_\Omega |F(x, y_0) - F(x_0, y_0)|^2 \mathrm{d}x \leq \tag{33}$$

$$\leq 4E_s^2 \int_\Omega |\Delta y_0|^2 \mathrm{d}x + 4\lambda ||f - f_0||_{\infty, \Omega} \int_\Omega \frac{1}{(1 - y_0)^2} \mathrm{d}x +$$

$$+ 4|\Omega|\chi^2 ||f||_{\infty, \Omega} \lambda \int_\Omega \frac{1}{1 - y_0} \mathrm{d}x.$$

However, being $y_0 \in Y$,

$$\int_\Omega |\Delta y_0|^2 \mathrm{d}x < M_1; \tag{34}$$

moreover, $1 - y_0 > d^*$, $\chi = \epsilon_0 L^2 (C_f d)^{-1}$ and $|\Omega| = L \cdot l$. Therefore, by (26),

$$||f - f_0||_{\infty, \Omega} = M_3 < +\infty \tag{35}$$

from which $||f||_{\infty, \Omega}^2 < M_4$, so that, also considering (3), (33) becomes

$$\int_\Omega |F(x, y_0) - F(x_0, y_0)|^2 \mathrm{d}x \leq 4 \left\{ E_s^2 M_1 + \frac{\epsilon_0 V^2 L^3 l}{2d^3 T d^*} \left[ \frac{M_3}{d^*} + \frac{L^3 l M_4}{C_f d} \right] \right\} = \tag{36}$$

$$= C_1 + C_2 V^2 \ll 1,$$

where (once the geometry of the device and the material constituting the deformable plate have been chosen) $C_1$ and $C_2$ are constants. This means that in these cases, the continuity of $x \to F(x, y_0)$, is voltage controlled, highlighting that, in the reference energy state, there are no appreciable anomalous phenomena of curvature of the deformable plate. This is in line with the experimental and industrial experience according to which the deformable plate does not show evident deformations (and/or curvatures) in the start-up phase [37,38].

*5.3. On the Injectibility of B On Y*

The deformable plate structurally responds to the theory of Sophie-Germain Lagrange [26,37,38] according to which, qualitatively,

$$\Delta^2 u(x) = \frac{p(x)}{D}, \tag{37}$$

where $p(x)$ is the mechanical pressure, $\Delta^2 = B$, $u(x)$ are admissible profiles of the deformable plate ($u(x) \in Y$) and $D$ flexural stiffness of the plate, is

$$D = \frac{Et^3}{12(1 - \nu)} \tag{38}$$

in which $t$ is the thickness of the deformable plate and $E$ and $\nu$ are the Young and Poisson modules, respectively. Then, due to both the uniqueness of $\Delta^2$ and its injectivity on $Y$, it

is legitimate to state that, given the load $p(x)$, the distribution of the deformations of the plate is completely defined. Then, by means of constitutive laws [26], it is possible to obtain $T(x)$ (mechanical tension of the deformable plate which, however, is a bounded function, and therefore increased by a constant) particularly useful for the choice of the material constituting the deformable plate.

*5.4. An Interesting Limitation for Applied Voltage V*

It seems only right to specify that in (20)

$$C(\beta, \lambda, \gamma, \delta, \chi, M, M_1, M_2, f)d_\Omega^{\frac{N}{2}} \ll 1 \tag{39}$$

(because $d_\Omega \ll 1$, from which $d_\Omega^{\frac{N}{2}} \ll 1$), so that

$$C(\beta, \lambda, \gamma, \delta, \chi, M, M_1, M_2, f) < d_\Omega^{-\frac{N}{2}}. \tag{40}$$

In addition, being $d_\Omega \ll 1$, it follows that $d_\Omega^{-\frac{N}{2}} \gg 1$, from which

$$C(\beta, \lambda, \gamma, \delta, \chi, M, M_1, M_2, f) \in [0, d_\Omega^{-\frac{N}{2}}) \tag{41}$$

where $[0, d_\Omega^{-\frac{N}{2}})$ is a sufficiently wide range of values capable of verifying the (20) for a variety of devices.

In [3], $C(\beta, \lambda, \gamma, \delta, \chi, M, M_1, M_2, f)$ has not been made explicit (limiting its indication in terms of functional dependence). In this paper, retracing each step of the proof of the Theorem 1, we obtain:

$$C(\beta, \lambda, \gamma, \delta, \chi, M, M_1, M_2, f) = \tag{42}$$

$$= d_\Omega^6 \left\{ 4M_1^2 \beta_2 + 576\lambda \|f\|_{\infty,\Omega}^2 \sqrt[8]{M^3}(d^{*2}\sqrt{M^8} + \chi^6)D_\Omega^{6N+2} + 16\lambda\delta^2 M_2 d_\Omega^{-\frac{N}{2}} < 1$$

from which, considering both (3) and (39), focusing our attention on 3$D$ formulation ($N = 3$), and selecting numerical values for the geometric parameters typical of many industrial applications ($d^* \to 0$ and $d_\Omega = \sqrt{3}$ [1,25,38]), we achieve the following limitation for $V$:

$$V < \sqrt{\frac{T\left(10^{33} - \frac{19.68 \cdot 10^{24} f^4}{T^4 d^4}\right)}{2549 \cdot 10^{-28} \sqrt[8]{\frac{10.94 \cdot 10^{36} f^6}{d^6}} + 17.8 \cdot \sqrt{3}M_2}} \tag{43}$$

where $\delta$ was fixed equal to 2 (worst case, as proved in [17]). Finally, exploiting (26), (43) becomes

$$V < \sqrt{\frac{T\left(10^{33} - \frac{19.68 \cdot 10^{24} |x|^{4\alpha}}{T^4 d^4}\right)}{2549 \cdot 10^{-28} \sqrt[8]{\frac{10.94 \cdot 10^{36} |x|^{6\alpha}}{d^6}} + 17.8 \cdot \sqrt{3}}}. \tag{44}$$

Finally, setting $|x| \le 0.5$ and $d = 11$, $\alpha = 2$, (44) becomes

$$V < 23.7 \cdot 10^7 \sqrt{10^{21}T - 19.68} \tag{45}$$

from which

$$T > 10^{-21}(18 \cdot 10^{-10}V^2 + 19.68). \tag{46}$$

**Remark 7.** *It is worth noting that the decimal numbers obtained in both (45) and (46) are the result of numerical approximations.*

**Remark 8.** *Figure 2 offers a graphical representation of (46) (where both T and V are no longer dimensionless), differentiating the areas allowed from those forbidden. Particularly, it is evident that the studied device allows a high number of intended uses ($V \in [0, 100]$ Volt), allowing the employment of a wide range of materials for the deformable plate ($T \in [1.968 \times 10^{-20}, +\infty$) $N/m^2$).*

**Remark 9.** *It is worth noting that the maximum allowed voltage, as obtained in (44), is strongly dependent on the electromechanical properties of the material constituting the deformable plate.*

**Remark 10.** *To achieve (44), $\chi \to 1$ to take into account possible incipient breakage of the device (bifurcation). Moreover, as specified in [17], all constants ($M$, $M_1$ and $M_2$) were set to 1.*

**Remark 11.** *Equation (1) does not allow explicit obtaining of solutions. Therefore, to obtain approximate solutions satisfying the analytical model (no ghost solutions), we rely on a numerical procedure which is considered the "gold standard" for this type of model.*

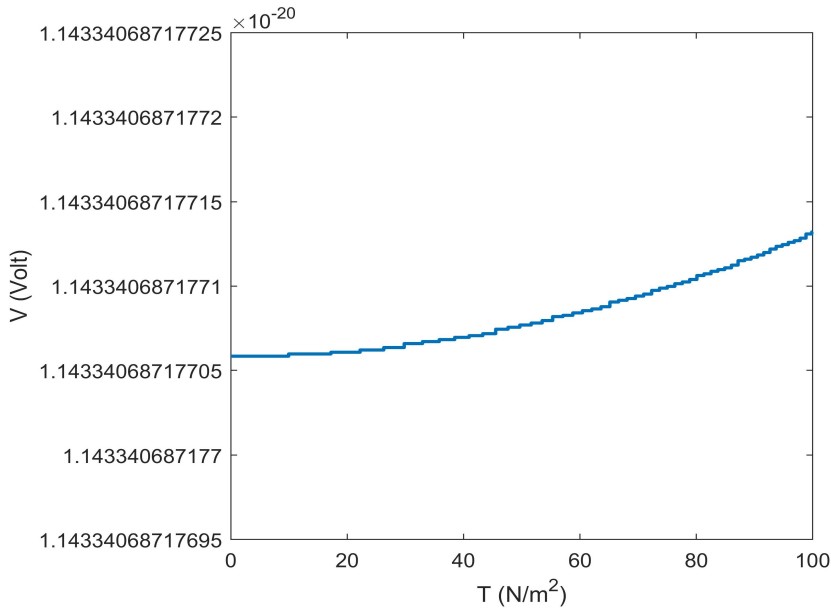

**Figure 2.** A representation of (46) when *V* changes: in it both the forbidden and permitted areas are indicated as regards the values of *V*.

## 6. How Fringing Field Affects Electrostatic Force

It is known that [17],

$$|\mathbf{E}|^2 = \frac{\epsilon_0 L^2 V^2}{2\theta d^3 T(1 - u(x))^2}, \quad \theta \in \mathbb{R}^+ \tag{47}$$

so that the electrostatic force, $f_{el}$, becomes:

$$f_{el} = \frac{\epsilon_0^2 L^3 V^2}{4\theta d^3 T d^3 (1 - u(x))^2}. \tag{48}$$

Moreover, if $\mathbf{E}_{fringing\ field}$ is the fringing field, one achieves

$$\frac{\lambda \delta |\nabla u(x)|^2}{(1 - u(x))^2} = \eta |\mathbf{E}_{fringing\ field}|^2, \quad \eta \in \mathbb{R}^2. \tag{49}$$

Therefore, the related electrostatic force becomes

$$[f_{el}]_{\mathbf{E}_{fringing\ field}} = \frac{0.5\lambda\delta|\nabla u(x)|^2}{(1 - u(x))^2} \tag{50}$$

so that, by (3), one easily achieves

$$[f_{el}]_{\mathbf{E}_{fringing\ field}} = \frac{\epsilon_0 \delta V^2 |\nabla u(x)|^2}{T(1 - u(x))^2}\left(\frac{L}{d}\right)^3. \tag{51}$$

Therefore, the total electrostatic force becomes:

$$(f_{el})_{TOT} = f_{el} + [f_{el}]_{\mathbf{E}_{fringing\ field}} = \tag{52}$$

$$= \left[\frac{\epsilon_0^2}{4\theta d^3(1 - u(x))^2} + \frac{\epsilon_0\delta|\nabla u(x)|^2}{(1 - u(x))^2}\right]\left(\frac{V^2}{T}\right)\left(\frac{L}{d}\right)^3$$

confirming that the $(f_{el})_{TOT}$ inside the device depends not only on $\delta$ but also on $L/d$ (algebraic ratio indicative of the possible presence of the fringing field). Here, $(f_{el})_{TOT}$ also depends on both $x$ and $T$ so that, increasing $T$, $(f_{el})_{TOT}$ decreases as required physically. To further confirm the applicability of (52) in industrial applications [39,40], when increasing $V$, $(f_{el})_{TOT}$ also increases. We finally note that, since $\frac{1}{(1-u(x))^2} < \frac{1}{d^{*2}}$, and in our case (no longer dimensionless) $d^* \approx 10^{-9}$ and $L \approx 10^{-6}$, from (52), we achieve a limitation for $(f_{el})_{TOT}$

$$(f_{el})_{TOT} < (24.12 \cdot 10^{38} + 21.14 \cdot 10^{-24}\delta)\frac{V^2}{T} \tag{53}$$

highlighting, once again, that the increase of $T$ causes a decrease in the effects due to the fringing field.

**Remark 12.** *According to Section 5.3, $T(x)$ is a bounded function such that $T = sup_x\{T(x)\}$, so that (53) is valid.*

**Remark 13.** *While it appears that (44) does not depend on $\delta$, (53) ensures that $T$ limits $(f_{el})_{TOT}$ where $\delta$ exists.*

**Remark 14.** *Both (52) and (53) confirm the important experimental issue according to which the effect due to $\mathbf{E}_{fringing\ field}$ depends on $L/d$. Particularly, if $L/d \ll 1$, the effects due to the $\mathbf{E}_{fringing\ field}$ will be significantly reduced (as experimentally verified in [41,42]).*

## 7. Deformable Plate Profile Recovering: A Numerical Approach

Equation (1), due to the way it is structured, is not suitable for numerical processing. Therefore, some simplifications in compliance with the conditions of global existence and uniqueness must be made.

### 7.1. Some Due Simplifications of the Analytical Model

From (1), taking into account (8) and considering that [17]

$$V = \frac{V_s}{1 + \chi \int_\Omega \frac{dx}{1-u(x)}}, \tag{54}$$

we can easily achieve

$$\int_\Omega \frac{dx}{1 - u(x)} = \frac{V_s - 1}{\chi} \tag{55}$$

so that (1) becomes

$$
\begin{cases}
\Delta^2 u(x) = \left( \beta \int_\Omega |\nabla u(x)|^2 \mathrm{d}x + \gamma \right) \Delta u(x) + \frac{\lambda f(x)}{(1-u(x))^2 (V_s)^2} + \lambda \delta |\nabla u(x)|^2 \\
u(x) = 0, \quad \nabla u(x) = 0 \quad x \in \partial \Omega, \\
0 < u(x) < 1 \quad x \in \Omega \subset \mathbb{R}^N, \quad N < 4.
\end{cases}
\tag{56}
$$

Again, setting $\frac{1}{V_s^2} = K_1$ (constant), we achieve

$$
\begin{cases}
\Delta^2 u(x) = \left( \beta \int_\Omega |\nabla u(x)|^2 \mathrm{d}x + \gamma \right) \Delta u(x) + \frac{\lambda K_1 f(x)}{(1-u(x))^2} + \lambda \delta |\nabla u(x)|^2 \\
u(x) = 0, \quad \nabla u(x) = 0 \quad x \in \partial \Omega, \\
0 < u(x) < 1 \quad x \in \Omega \subset \mathbb{R}^N, \quad N < 4.
\end{cases}
\tag{57}
$$

Assuming that $\gamma$ takes precedence over $\beta$, (57) takes its final form:

$$
\begin{cases}
\Delta^2 u(x) = \gamma \Delta u(x) + \frac{\lambda K_1 f(x)}{(1-u(x))^2} + \lambda \delta |\nabla u(x)|^2 \\
u(x) = 0, \quad \nabla u(x) = 0 \quad x \in \partial \Omega, \\
0 < u(x) < 1 \quad x \in \Omega \subset \mathbb{R}^N, \quad N < 4
\end{cases}
\tag{58}
$$

which represents the simplified version of (1) numerically implementable by the finite difference approach (the "gold standard" procedure for this type of problem which enables us to avoid the risk of approximate solutions representing ghost solutions).

**Remark 15.** *We observe that (58), even if it is a simplified version of (1), does not invalidate the verification of all the properties tested in the previous sections (we omit the proof for reasons of space, as it retraces the five steps of the proof of Theorem 1 [17]).*

### 7.2. Derivation of the Numerical Model: Finite Difference Approach

In this section, we derive the finite difference method for the non linear boundary values problems (58) in $2D$ space. In particular:

$$
\begin{cases}
\Delta^2 u(x) = \gamma \Delta u(x) + \frac{\lambda K_1 f(x)}{(1-u(x))^2} + \lambda \delta (v(x)^2 + w(x)^2) \\
u(x) = 0, \quad v(x) = 0, \quad w(x) = 0, \quad x \in \partial \Omega \\
0 < u(x) < 1 \quad x \in \Omega \subset \mathbb{R}^2,
\end{cases}
\tag{59}
$$

where we assume $v(x) = u_x(x)$ and $w(x) = u_y(x)$. We consider the computational domain $-1 \le x \le 1, -1 \le y \le 1$ and use a uniform Cartesian grid consisting of grid points $(x_i, y_j)$, where $x_i = i\Delta x$ and $y_j = j\Delta y$, for $i = 0, \cdots, I$ and $j = 0, \cdots, J$, with $\Delta x$ and $\Delta y$ increments in $x-$direction and $y-$direction, respectively. In this context, we consider the special case where $\Delta x = \Delta y = h$, so that $I = J = \overline{N}$ number of intervals in both directions.

Let $U_{ij}$ be the approximation to the exact solution $u(x_i, y_j)$ at the grid points $(x_i, y_j)$. To discretize the model (59) at each grid point $(x_i, y_j)$ of the computational domain, we use the following second order centered finite differences

$$
\begin{aligned}
u_{xx} &\approx \frac{U_{i+1,j} - 2U_{ij} + U_{i-1,j}}{h^2}, \\
u_{yy} &\approx \frac{U_{i,j+1} - 2U_{ij} + U_{i,j-1}}{h^2}, \\
u_x &\approx \frac{U_{i+1,j} - U_{i-1,j}}{2h} \\
u_y &\approx \frac{U_{i,j+1} - U_{i,j-1}}{2h}.
\end{aligned}
$$

Let $V_{ij}$ and $W_{ij}$ be the approximations to the $x-$ and $y-$derivatives $u_x(x_i, y_j)$ and $u_y(x_i, y_j)$ of the exact solution, respectively; thus, we obtain the finite difference method

$$
\begin{aligned}
&s_2 U_{i-1,j} + s_1 U_{i+1,j} - 4U_{ij} + s_2 U_{i,j-1} + s_1 U_{i,j+1} = h^2 F(U_{ij}) \\
&U_{i+1,j} - U_{i-1,j} - 2hV_{ij} = 0 \\
&U_{i,j+1} - U_{i,j-1} - 2hW_{ij} = 0,
\end{aligned}
\tag{60}
$$

for $i, j = 1, \cdots, \overline{N} - 1$, with $s_1 = 1 - 0.5h\gamma$ and $s_2 = 1 + 0.5h\gamma$ and where

$$
F(U_{ij}) = \frac{\lambda K_1 f(x_i, y_j)}{(1 - U_{ij})^2} + \lambda\delta(V_{ij}^2 + W_{ij}^2).
\tag{61}
$$

The finite difference Equations (60) at points near the boundary involve the known boundary values, which, are generally moved to the right-hand side. However, the zero boundary conditions do not contribute to the model under study. We thus obtain a system of $3(\overline{N} - 1)^2$ nonlinear equations in $3(\overline{N} - 1)^2$ unknowns, $U_{ij}, V_{ij}$ and $W_{ij}$, that can be expressed in the vector form

$$
AU = F(U)
\tag{62}
$$

where $U$ is the vector of the unknowns

$$
U = \begin{bmatrix} U \\ V \\ W \end{bmatrix}
\tag{63}
$$

with $U, V, W$ assigned by the natural row-wise ordering

$$
Z = \begin{bmatrix} Z^{[1]} \\ Z^{[2]} \\ \vdots \\ Z^{[N-1]} \end{bmatrix}
\quad \text{where} \quad
Z^{[j]} = \begin{bmatrix} Z_{1j} \\ Z_{2j} \\ \vdots \\ Z_{N-1j} \end{bmatrix}
\quad \text{for} \quad j = 1, \cdots, N-1,
$$

with $Z = U, V, W$.

The $A$ matrix is a very sparse square of size $3(\overline{N} - 1)^2 \times 3(\overline{N} - 1)^2$

$$
A = \begin{bmatrix} A_1 & A_4 & A_7 \\ A_2 & A_5 & A_8 \\ A_3 & A_6 & A_9 \end{bmatrix}
\tag{64}
$$

with a block structure as shown in Figure 3. Since each finite difference equation involves at most five unknowns, each row of the matrix $A$ has at most five non-zeros elements.

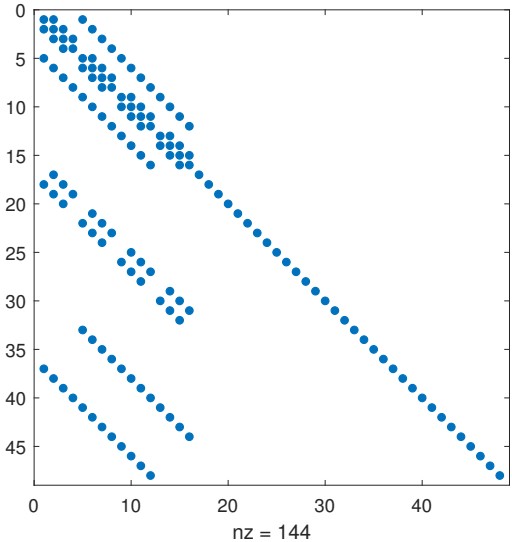

**Figure 3.** Example of the $A$ matrix for $\overline{N} = 5$.

Each sub-matrix $A_k$, $k = 1, \cdots, 9$, has a block form

$$A_k = \begin{bmatrix} T & S_1 & & & \\ S_2 & T & S_1 & & \\ & S_2 & T & S_1 & \\ & & \ddots & \ddots & \ddots \\ & & & S_2 & T \end{bmatrix} \tag{65}$$

where each block, $T$, $S_1$ and $S_2$, is a $(\overline{N} - 1) \times (\overline{N} - 1)$ matrix.

In particular, for the sub-matrix $A_1$, $T$ has a tridiagonal form

$$T = \begin{bmatrix} -4 & s_1 & & & \\ s_2 & -4 & s_1 & & \\ & s_2 & -4 & s_1 & \\ & & \ddots & \ddots & \ddots \\ & & & s_2 & -4 \end{bmatrix} \tag{66}$$

with $s_1 = 1 - 0.5h\gamma$ and $s_2 = 1 + 0.5h\gamma$. $S_1$ and $S_2$ are diagonal matrices with elements equal to $s_1$ and $s_2$, respectively. The non-zeros values of the matrices $S_1$ and $S_2$ are separated from the diagonal by $\overline{N} - 1$ zeros, since these coefficients correspond to grid points lying above or below the central point in the stencil, and are hence in the next or previous row of unknowns. For the sub-matrix $A_2$, it has

$$T = \begin{bmatrix} 0 & s_1 & & & \\ s_2 & 0 & s_1 & & \\ & s_2 & 0 & s_1 & \\ & & \ddots & \ddots & \ddots \\ & & & s_2 & 0 \end{bmatrix} \tag{67}$$

with $s_1 = -1$ and $s_2 = 1$. The matrices $S_1$ and $S_2$ are null. For the sub-matrix $A_3$, it has $T$ null matrix and $S_1$ and $S_2$ diagonal matrices with element equal to $s_1 = 1$ and $s_2 = -1$.

The sub-matrices $A_5$ and $A_9$ are diagonal, with principal elements equal to $-2h$; the other sub-matrices are null. Next, the *fsolve.m* Matlab®R2019a routine, running on an Intel Core 2 CPU at 1.45 GHz , is used to solve the nonlinear algebraic system (62), with initial guess $U_0 = 0$.

The numerical results reported in Figure 4 are obtained by setting $\gamma = 0.1$, $\lambda = -0.1$, $\delta = 0.2$ and $K_1 = 0.9$ with $f(x) = |x|^{0.2}$ and $h = 0.05$.

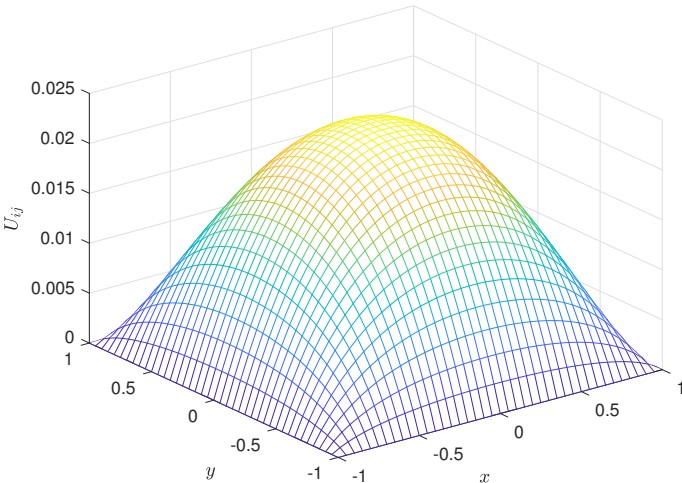

**Figure 4.** Recovering of the deformable plate profile when $\gamma = 0.1$, $\lambda = -0.1$, $\delta = 0.2$ and $K_1 = 0.9$ with $f(x) = |x|^{0.2}$ and $h = 0.05$.

**Remark 16.** *We observe that the obtained matrix is sparse and (at times) large, requiring the use of iterative techniques for solving linear systems based on Krylov sub-spaces ([43]). However, in this paper, the dimensions of the array are not such as to require the imperative use of these methods. Furthermore, since the application under study is of an off-line type, the calculation times performed by fsolve.m do not affect the validity of the proposed procedure.*

**Remark 17.** *The finite difference method approximates the value of the derivative of a function in a point (for which it would be necessary to know all the values of the function (therefore infinite) in a neighborhood of the same point), with an expression that takes them into account only a finite number (often very small). This is passed from the limit operation to the incremental ratio operation. This allows to transform a differential equation into an algebraic problem whose sparse matrix strongly depends on the number of values used in the approximation of the derivatives. Starting from the considerations on the orders of accuracy of the derivative approximation formulas, it is easy to prove that the error made by approximating the second derivative is a $\mathcal{O}(h^2)$. Moreover, the approximation error between the true solution $u(x)$ of the problem (1) and the approximate one is of the second order. In fact, introducing the truncation error $\tau_j$, and exploiting the Taylor series developments, it is easy to achieve:*

$$||\tau_j|| = \frac{h^2}{12}||g''(x_i)|| = \mathcal{O}(h^2) \tag{68}$$

*where g must be at least $C^2$. Moreover, introducing the vector of the global error, $\mathbf{H}$, we can easily achieve:*

$$||\mathbf{H}|| \leq ||A^{-1}|| \, ||\tau|| \tag{69}$$

*and again $||\tau|| \to 0$ (which ensure the consistency of the numerical approach) and $||A^{-1}|| \leq C$, with C independent on h (which ensure the stability of the numerical approach).*

## 8. Some Interesting Numerical Results

*Approximate $u(x)$ and Ghost Solutions*

Equations (19), (20) and (21) depict the conditions which numerical solutions must satisfy to avoid representing ghost solutions. Once implemented, the numerical procedure was tested for several experimental cases intended for industrial applications. Particularly, assuming $f(x) = |x|^{0.2}$, $h = 0.05$, $K_1 = 0.9$ [33], max $U_{ij}$ have been achieved when $\delta$ increases (see Table 2 where it is also highlighted that max $U_{ij}$ satisfy (19), (20) and (21)).

The max $U_{ij}$ obtained shows that the hypothesis of small displacements is also confirmed in the numerical recovering of the deformable plate profile, with a slight increase due to the growth in the effects caused by the fringing field. The results shown in Table 2 refer to a deformable plate with $\gamma = 0.1$; however, different values of $\gamma$ did not produce significant changes in max $U_{ij}$ with respect to the case $\gamma = 0.1$. This is because $\gamma$, in (1), has little impact, compared to the other addend, in weighing $\Delta u(x)$ (which governs the curvature of the deformable plate). Moreover, as $\delta$ increases, the contribution due to $V$ increases, thus even high values of $V$ (19), (20) and (21) are verified. In addition, the effects due to $\delta$ affect $\Delta^2 u(x)$: there is a prevalence of the effect due to $\delta$ as sanctioned by (1), because $u(x)$ increases its concavity when $\delta$ increases ($\mathbf{E}_{\text{fringing field}}$ favors the deformation of the deformable plate). However, $|\nabla u(x)|^2$ in (1), has little effect numerically (for implementation reasons), therefore the underestimation of max $U_{ij}$ could be compensated for by amplifying factors to be researched experimentally.

We observe that the numerically constructed regularity of $u(x)$ suggests that the performance of the approach is satisfactory even if the high computational complexity (with respect to other numerical approaches, [44]) would make the approach inconvenient for any real-time industrial applications (i.e., when short plate recovery times are required). However, these applications are currently not widespread on a large scale.

**Table 2.** max $U_{ij}$ when $f(x) = |x|^{0.2}$, $h = 0.05$, $K_1 = 0.9$.

| $\delta$ | $\gamma$ | $\lambda$ | **max $U_{ij}$** | (19) | (20) | (21) |
|---|---|---|---|---|---|---|
| 0.2 | 0.1 | −0.1 | 0.023028788820237 | verified | verified | verified |
| 0.2 | 0.3 | −0.1 | 0.023000926902868 | verified | verified | verified |
| 0.2 | 0.5 | −0.1 | 0.022920240529700 | verified | verified | verified |
| 0.2 | 0.1 | −0.2 | 0.047876393295838 | verified | verified | verified |
| 0.2 | 0.3 | −0.2 | 0.047808989792072 | verified | verified | verified |
| 0.2 | 0.5 | −0.2 | 0.047625803377756 | verified | verified | verified |
| 0.2 | 0.1 | −0.5 | 0.139135198053765 | verified | verified | verified |
| 0.2 | 0.3 | −0.5 | 0.138790292277593 | verified | verified | verified |
| 0.2 | 0.5 | −0.5 | 0.138011485983744 | verified | verified | verified |
| 0.2 | 0.1 | −0.8 | 0.300973743344764 | verified | verified | verified |
| 0.2 | 0.3 | −0.8 | 0.298510411698396 | verified | verified | verified |
| 0.2 | 0.5 | −0.8 | 0.293792489940541 | verified | verified | verified |

Figures 5–9 show some recovering of deformable plate profile as the amplitude of $\lambda$ increases. From these profiles it can be deduced once again how, as this amplitude increases (and therefore as the externally applied $V$ increases) the profile of the deformable plate rises more and more. Obviously, the values of max $U_{ij}$ are not such as to fear the risk of the deformable plate touching the top plate of the device.

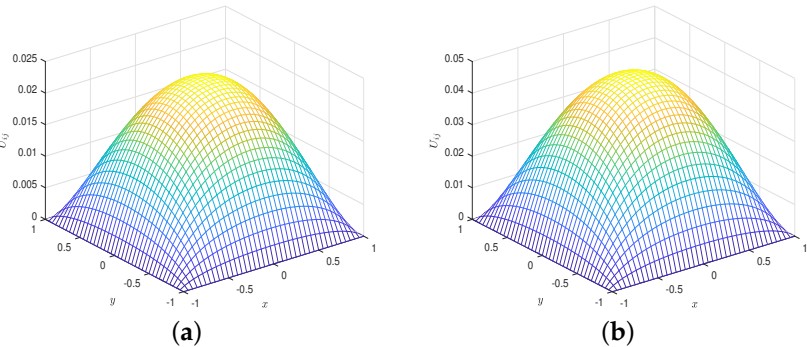

(a)  (b)

**Figure 5.** Recovering of the deformable plate profile for $\delta = 0.2$ and $K_1 = 0.9$ with $f(x) = |x|^{0.2}$ and $h = 0.05$ when (**a**) $\gamma = 0.3$, $\lambda = −0.1$ and (**b**) $\gamma = 0.1$, $\lambda = −0.2$.

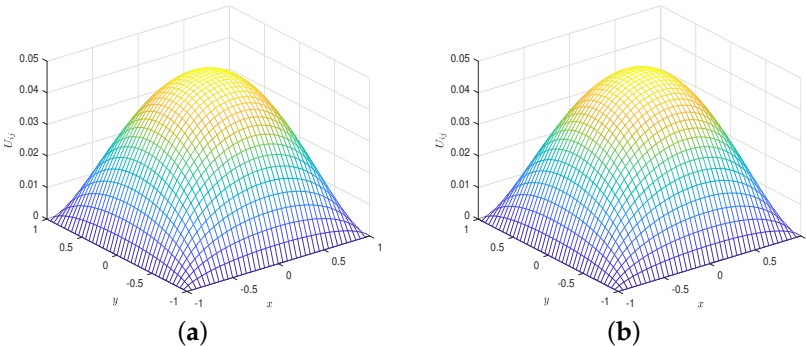

**Figure 6.** Recovering of the deformable plate profile for $\delta = 0.2$ and $K_1 = 0.9$ with $f(x) = |x|^{0.2}$ and $h = 0.05$ when (**a**) $\gamma = 0.3$, $\lambda = -0.2$ and (**b**) $\gamma = 0.5$, $\lambda = -0.2$.

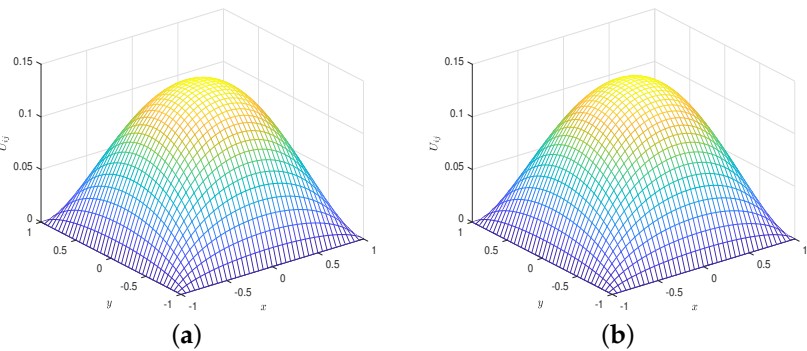

**Figure 7.** Recovering of the deformable plate profile for $\delta = 0.2$ and $K_1 = 0.9$ with $f(x) = |x|^{0.2}$ and $h = 0.05$ when (**a**) $\gamma = 0.1$, $\lambda = -0.5$ and (**b**) $\gamma = 0.3$, $\lambda = -0.5$.

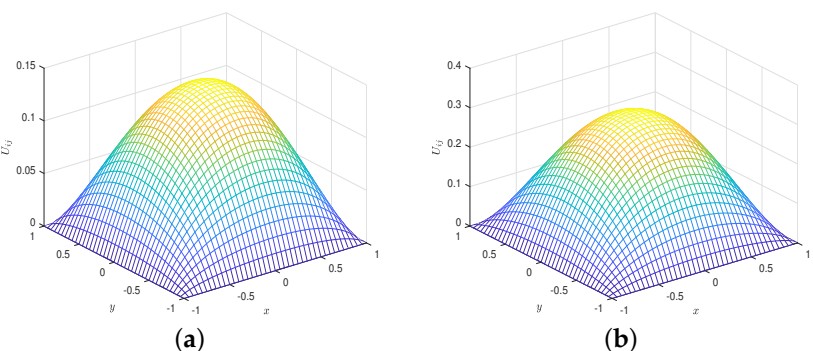

**Figure 8.** Recovering of the deformable plate profile for $\delta = 0.2$ and $K_1 = 0.9$ with $f(x) = |x|^{0.2}$ and $h = 0.05$ when (**a**) $\gamma = 0.5$, $\lambda = -0.5$ and (**b**) $\gamma = 0.1$, $\lambda = -0.8$.

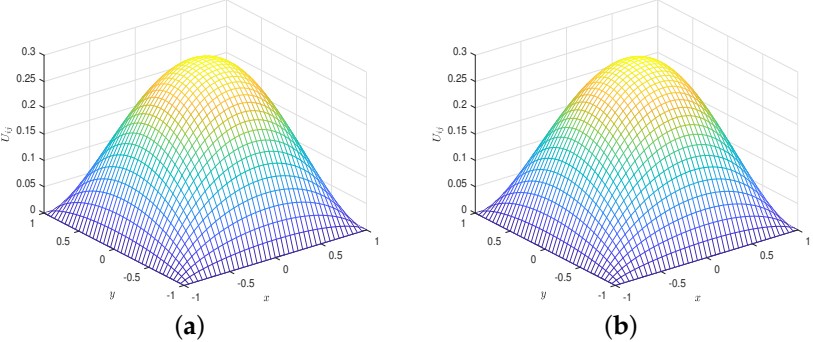

**Figure 9.** Recovering of the deformable plate profile for $\delta = 0.2$ and $K_1 = 0.9$ with $f(x) = |x|^{0.2}$ and $h = 0.05$ when (**a**) $\gamma = 0.3$, $\lambda = -0.8$ and (**b**) $\gamma = 0.5$, $\lambda = -0.8$.

**Remark 18.** *It is worth noting that in* (1) *λ is a parameter which, apparently, would turn out to be positive. However, the numerical recovering of the deformable plate profile required negative values of λ. This is due to the fact that λ > 0 would derive from an external V inversely polarizing the device.*

## 9. A Further Limitation for $T$

The following results yield:

**Proposition 1.** *For* (1), *the following limitation for T holds:*

$$10^{-21}(18 \cdot 10^{-10} V^2 + 19.68) < T < \frac{1}{K_1 f(x)} \left( \frac{1}{\theta} + \frac{1}{\eta} \delta H \right) \frac{\epsilon_0^2 V^4 L^4}{8 d^9} \tag{70}$$

**Proof.** From (57), we achieve

$$\lambda \left( \frac{K_1 f(x)}{(1 - u(x))^2} + \delta |\nabla u(x)|^2 \right) = \left( \theta |\mathbf{E}|^2 + \eta |\mathbf{E}_{fringing field}|^2 \right), \tag{71}$$

from which, by (3), (47) and (49), we achieve

$$\theta \lambda = \frac{\epsilon_0^2 V^4 L^4}{4 d^6 T^2} \left( \frac{K_1 f(x)}{(1 - u(x))^2} + \delta |\nabla u(x)|^2 \right) \frac{1}{\left( \frac{1}{\theta} \frac{\epsilon_0 L^2 V^2}{2 d^3 T (1 - u(x))^2} + \frac{1}{\eta} \frac{\epsilon_0 V^2 L^2 \delta |\nabla(x)|^2}{2 d^3 T (1 - u(x))^2} \right)}. \tag{72}$$

From (72), considering that $|\nabla u(x)|^2$ is bounded [17], therefore $|\nabla u(x)|^2 < H$ ($H$ constant). Moreover, being $u(x) < 1$, it is easy to achieve

$$T < \frac{1}{K_1 f(x)} \left( \frac{1}{\theta} + \frac{1}{\eta} \delta H \right) \frac{\epsilon_0^2 V^4 L^4}{8 d^9} < H_1 V^2 \tag{73}$$

with $H_1 = \frac{1}{K_1 f(x)} \left( \frac{1}{\theta} + \frac{1}{\eta} \delta H \right)$ constant (because $f(x)$ is a bounded function). Therefore, by (46) and (73), we achieve (70), which represents a good limitation for $T$, depending on both $x$ (due to $f(x)$) and $\delta$. Moreover, (73) also depends on $|\mathbf{E}|$ (by $\theta$). In addition, fixed $L$, plates with higher $T$ can be destined in MEMS with plates very close to each other (reduced values of $d$), because high values of $T$ force the deformable plate not to touch the plate superior. On the contrary, if fixed $d$, deformable plates with reduced $T$ can be exploited in MEMS with reduced value of $L$.  □

**Remark 19.** *We observe that $d^9$ (with $d = 10^{-9}$) in* (70) *allows us to validate Remark 8.*

**Remark 20.** *Figure 10 displays the link between T and V as governed by* (73). *Therefore, as qualitatively deduced in [27], $T = KV^2$, so that once the intended use of the MEMS has been established, the straight line parallel to the ordinate axis (passing through $(V, 0)$) intercepts the curve at a point, in this case, all the points below it select T eligible. As in [27], Figure 10 can be employed to choose the intended use, beginning with the selected material.*

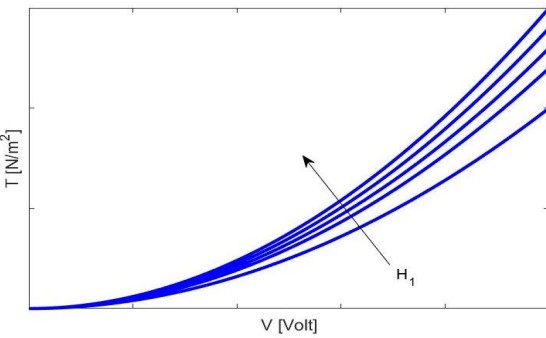

**Figure 10.** $T$ and $V$: an important link.

## 10. On the Influence of the Properties of the Deformable Plate on the Proposed Procedure

As verified, obtaining the limitations for $T$ and $V$ depend on the electro-mechanical properties of the material constituting the deformable plate (presence of both $\theta$ and $\lambda$) influencing the concavity of $u(x)$, highlighting that the increments of $T$ reduce the amplitudes of $u(x)$. On the other hand, the increase of $T$ produces smaller deformations at the edges. Furthermore, the numerical approach depends on the above parameters which, in a certain sense, govern the convergence. Finally, as in [27], these electro-mechanical properties influence the choice of $T$ starting from $V$, and vice versa. Particularly, the presence of $T$ in the denominator in the limitation of $V$ strongly influences $(f_{el})_{TOT}$, and therefore $|\mathbf{E}|$ total which, through (1), affects $\Delta^2 u(x)$ (higher order curvatures of the deformable plate); thus, the greater the $T$, the lower the concavity of the deformable plate (also confirming that plates that are more mechanically resistant show reduced deformations at the edges). Furthermore, $\lambda$ in (19), (20), and (21) ensure that $T$ also influences the algebraic conditions of existence and uniqueness of the solution for (1).

We highlight that the numeric procedure depends on $\lambda$, $\gamma$, and $\delta$, which govern the convergence of the method obtaining $u(x)$ which satisfy (19), (20), and (21). Finally, by (53), it becomes clear that both $V$ and $T$ act effectively on $(f_{el})_{TOT}$. In other words, both the intended use of the device and the mechanical properties of the deformable plate affect the behavior of the device. To summarize, the model studied is more versatile than other simplified models where the intended use of the device imposed $T$ and vice versa.

## 11. Possible Intended Uses of the MEMS Under Study

Industrially, the device presented here does not require particular construction requirements, and therefore the production costs would be low. Furthermore, both (45) and (46) ensure that the device is usable for a wide range of industrial applications (i.e., biomedical applications, micro-pumps for intravenous drug administration, and surgical micro-systems). Obviously, the exact value of $V$ strictly determines the intended use of the device.

Nowadays, parallel plate devices are industrially widespread, especially when their use is integrated in further electro-mechanical devices. However, even if the electronic tests on MEMS have reached high levels of reliability, the mechanical tests still do not present similarly high levels of performance. Despite this, in recent years, the joint use of electronic and mechanical components mounted on a single chip has allowed very high performances of micro-sensors used in robotics and electronics for use, for example, in large metropolitan areas/Smart Cities.

## 12. Concerning the Pull-In Voltage and Electrostatic Pressure

In MEMS, applying $V$, is its deformable element. However, high values of $V$ (beyond the "pull-in" value) generates instability, because the electrostatic forces exceed the elastic ones (a potentially destructive phenomenon). In other words, from (3), if $\lambda > \lambda^*$ (with $\lambda^*$, pull-in value), (1) does not admit solutions. Conversely, if $\lambda < \lambda^*$ the solution for (1) exists.

As known in the literature, $\lambda^*$ has been proved both mathematically [1] and experimentally [45–47]. Particularly, following the approach proposed in [27], we plot the trends of $u_0$ (maximum deflection of the deformable plate) versus $\lambda$ (starting from $\lambda = 0$, there will be a value of $\lambda$, indicated with $\lambda^*$ below which there are two solutions while above there is no solution, thus $\lambda^*$ represents the bifurcation point). Figure 11 depicts the rends of $u_0 - \lambda$, both without fringing field ($\delta = 0$) and with it ($\delta = 1$), respectively. Both trends highlight the superimposition with experimentally obtained bifurcation diagrams [45]. In fact, in non dimensionless conditions, the pull-in voltage, $V_{pull-in}$, depending on the gap between the two plates, is displayed in Figure 12 whose trend is completely similar to those displayed in Figure 6 in [45] and Figure 6 in [48] where an experimental setup, consisting of two plates separated by a distance $d$ (each plate was acrylic sheet coated with conductive aluminum) highlighting that (1), even if referred to a simplified geometry, in terms of pull-in voltage, has a well-established experimental confirmation in the literature.

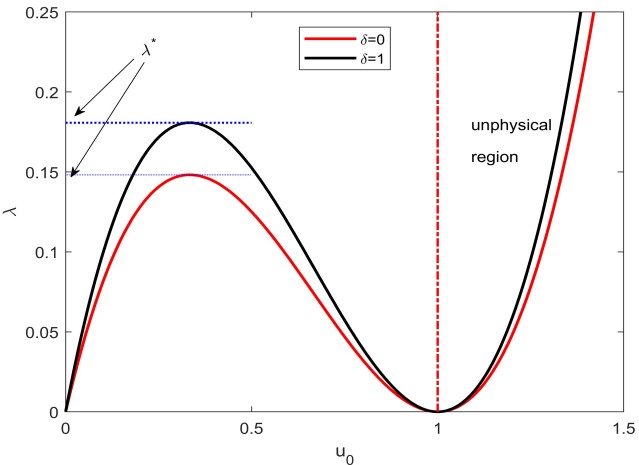

**Figure 11.** Dimensionless $\lambda^*$ as a function of dimensionless $u_0$.

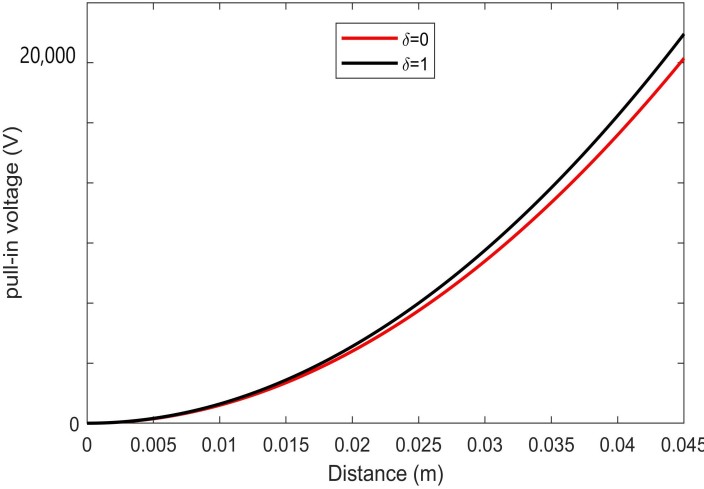

**Figure 12.** Pull-in voltage as function of distance $d$.

This important experimental finding is confirmed by the fact that, from (3), it is easy to achieve

$$V_{pull-in} = \sqrt{\frac{2d^3 T \lambda^*}{\epsilon_0 L^2}} \qquad (74)$$

which is completely analogous to (11) achieved in [45]. In addition, (74) is still analogous to (3) in [49] where a MEMS switch with perforated serpentine (Au) was designed, sim-

ulated, fabricated and characterized. Not last, (74) is quite similar to (1) in [48] where an electrostatic MEMS with parallel plates was considered.

We also note that, from (45), we can write:

$$V_{\text{max admissible}} = 23.7 \cdot 10^7 \sqrt{10^{21} T - 19.68}. \tag{75}$$

Therefore, considering the usual values for each parameter, we achieve $V_{pull\text{-}in} \approx V_{\text{max admissible}}$, with the consequence that the obtained $V_{\text{max admissible}}$ is compatible with the experimental setups known in the literature. Particularly, Table 3, after fixing the geometry of the device and varying the material constituting the deformable plate as in [49], reports the values of $V_{\text{max admissible}}$ achieved which result similar to those obtained in [49].

**Table 3.** Performance in terms of dimensions and materials.

| Material | Tickness (μm) | Length (μm) | Width (μm) | $V_{\text{max admissible}}$ (V) | $V_{\text{max admissible}}$ in [49] (V) | Deviation (V) |
|---|---|---|---|---|---|---|
| Au | 1.5 | 320 | 200 | 10.45 | 10.50 | 0.05 |
| Au | 1 | 320 | 200 | 8.12 | 8.20 | 0.08 |
| Au | 0.5 | 320 | 200 | 6.62 | 6.50 | −0.12 |
| Al | 1.5 | 320 | 200 | 7.12 | 7.10 | −0.02 |
| Al | 1 | 320 | 200 | 5.81 | 5.70 | −0.11 |
| Al | 0.5 | 320 | 200 | 3.77 | 3.85 | 0.008 |
| Cu | 1.5 | 320 | 200 | 8.23 | 8.45 | 0.22 |
| Cu | 1 | 320 | 200 | 6.15 | 6.25 | 0.10 |
| Cu | 0.5 | 320 | 200 | 4.28 | 4.39 | 0.11 |

As already highlighted, the effects due to the fringing field depend on the aspect ratio $L/d$. Then, the pull-in voltage is also affected by this relationship. Figures 13 and 14 highlight, for aspect ratio $L/d = 1, 2$, the trends of $V_{pull\text{-}in}$ using both the numerical procedure used in [45] and with the procedure proposed here. It should be noted that the $d$ increase produces a deviation from the trend obtained using [45], while agreeing with the experimental trend (as also highlighted in [45]). So, once again, we highlight the adherence of the theoretical results discussed in this paper (Deriving from (1)) with important experiments of electrostatic deflections.

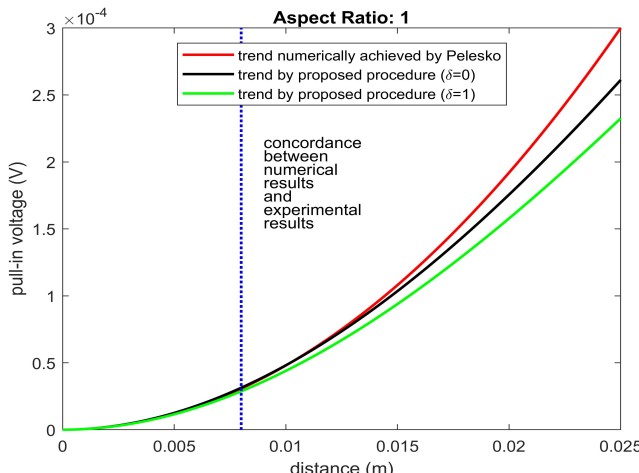

**Figure 13.** Trend of $V_{pull\text{-}in}$: $L/d = 1$. It should be noted that the agreement between numerical results and experimental findings already occurs from extremely small values of distance between the plates.

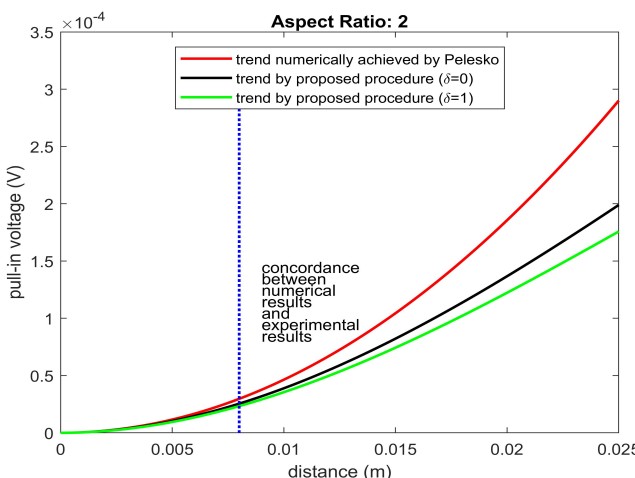

**Figure 14.** Trend of $V_{pull-in}$: $L/d = 2$. Even in the presence of intense effects due to the fringing field ($\delta = 2$), the concordance of the theoretical results discussed in this work with the experimental evidence is still evident.

Finally, from (51), the electrostatic pressure due to $\mathbf{E}_{fringing\ field}$ can be written as:

$$[p_{el}]_{\mathbf{E}_{fringing\ field}} = \frac{\epsilon_0 \delta V^2 |\nabla u(x)|^2}{T(1-u(x))^2} \frac{L^2}{d^3} \tag{76}$$

which, combined with (74), provide the link between $[p_{el}]_{\mathbf{E}_{fringing\ field}}$ and $V_{pull-in}$ that produce the following trend (as shown in Figure 15):

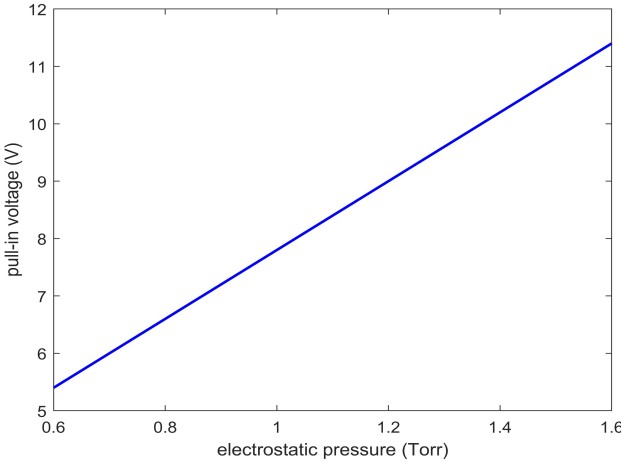

**Figure 15.** $V_{pull-in}$ versus $[p_{el}]_{\mathbf{E}_{fringing\ field}}$. The adherence of the theoretical result with the experimental evidence obtained in [50] is evident.

Quite similar to that obtained in [50], concluding once again that (1) is in line with many important experimental results already known in the literature.

**Remark 21.** *We observe that, as discussed in Section 12, the results obtained are mainly comparable with experimental setups obtained on electrostatic membrane MEMS devices. This is due to the fact that the global existence and uniqueness conditions for (1) have been obtained starting from (31) (reference energy state), which models an electrostatic membrane MEMS device with fringing field. Furthermore, the hypothesis of small displacements, together with the fact that β and γ in (1), do not vary during the deformation, the results for (1) agree with the experimental setups for membrane devices. In the near future, we hope to use β and γ variables to extend the comparison to further experimental setups.*

**Remark 22.** *Regarding the possible experimental confirmation of the results achieved for T,
Figure 12 depicts the link obtained between the pull-in voltage and the distance between the plates of
the device. This link was found to be superimposable to experimental links known in the literature.
Therefore, by (74), also T, indirectly, is verified experimentally.*

## 13. Conclusions and Perspectives

In this paper, a dimensionless fourth-order integro-differential model known in the
literature was considered while modeling the behavior of an electrostatic MEMS device
with metal parallel plates in which the effects due to the fringing field are modeled, ac-
cording to the Pelesko–Driscoll theory, by means of an easily implementable addend via
software/hardware whose weight can be controlled in voltage (parameter $\lambda$). This model
was formalized so that the device can be controlled in voltage by a suitable capacitive
control circuit. Starting from the conditions of global existence and uniqueness, the analyti-
cal results were discussed, highlighting the most important and captivating implications
for our industrial realities. In particular, the choice of a particular dielectric profile of
the deformable plate (satisfying mandatory physical requirements) led to obtaining im-
portant limitations for the external electrical voltage, for the mechanical tension of the
deformable plate, and for the total electrostatic force (including the contribution due to
the fringing field) highlighting relevant adherence with experimental evidence known
in the literature. It is undoubtedly worth noting that both the limitations for $V$ and for
$T$ allow you to reconstruct $u(x)$ profiles (model solutions) in complete safety. In other
words, once a particular device is fixed (i.e., by fixing $T$ of the material constituting the
deformable plate), the external $V$ applied satisfying the above limitations, it will allow
$u(x)$ profiles of the deformable plate complying with the safety standards according to
which the deformable element must not touch the upper wall of the device. Conversely,
by selecting the intended use of the device (i.e., $V$ satisfying the above limitations), it is
possible to choose a particular material for the deformable plate (whose $T$ satisfies the
relative limitation) capable of guaranteeing solutions $u(x)$ of high security. Since the model
does not allow the explicit recovering of the deformable plate, a numerical technique based
on finite differences (the "gold standard" for this particular type of analytical models) was
used to obtain profiles that are fully compatible with the conditions of global existence
and uniqueness. Furthermore, a simple criterion for choosing the intended use has been
provided, beginning with the mechanical tension of the deformable plate and vice versa,
which can be particularly useful for industrial applications. In particular, the high value of
$V_{\text{max admissible}}$ allows the device studied to be used also in intended uses where $V$ would
be high (industrial applications that require up to Class 3B micro devices (characterized
by high fault electrical voltage) according to the ESD/CEI classification). Furthermore,
the direct dependence of $V_{\text{max admissible}}$ on $T$ allows us to confirm what is required by the
industry according to which the device characterized by a deformable plate with high
rigidity allows uses with an even higher $V_{\text{max admissible}}$ (with profiles $u(x)$ such as not to
compromise the functionality of the device). The quality of the results obtained was also
confirmed by comparison with experimental setups known in the literature, highlighting
that the hypothesis of small displacements, together with the non-variability of some me-
chanical parameters during the deformation of the plate, allows an exhaustive comparison
of performing models of electrostatic devices membrane, which are famously easier to im-
plement. Thus, the possibility emerges of opening a new line of research into comparisons
between complete (but difficult to implement) models and simplified models representing
devices that are well-suited to experimental performances. We underline that the model,
being an ordinary derivative one, does not lend itself to being solved using FEM techniques
(particularly suitable for dynamic partial derivative models). Therefore, it is necessary to
reformulate the model in the near future so that its most important dynamic aspects are
considered (including the dependence on electrical conductivity and temperature), on one
hand, to obtain the more general conditions of global existence and uniqueness, and, on

the other, to proceed with the recovering of the deformable profile using FEM techniques (performing for software/hardware applications).

**Author Contributions:** Conceptualization, P.D.B., L.F. and M.V.; methodology, L.F. and M.V.; validation, P.D.B., A.J. and L.F.; formal analysis, L.F. and M.V.; investigation, P.D.B., A.J. and M.V.; resources, P.D.B., L.F. and M.V.; writing–original draft preparation, M.V.; writing–review and editing, P.D.B. and L.F.; supervision, P.D.B., L.F., A.J. and M.V. All authors have read and agreed to the published version of the manuscript.

**Funding:** This research received no external funding.

**Conflicts of Interest:** The authors declare no conflict of interest.

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
