# Peer review of "Finite Differences for Recovering the Plate Profile in Electrostatic MEMS with Fringing Field"

_electronics, doi:10.3390/electronics11193010_

Round 1

Reviewer 1 Report (New Reviewer)

Abstract looks more like Introduction. It should be rewritten.

There are too many self-references. Is it possible to reduce it? Why use self-citations in cases with more references [xx-xx]. I think that self-references before [26] are not necessary?

From the title, I expected more practical examples. You should modify the title to be better suited.

Aims of the paper should be rewritten. For aims, it is not usual to have "By carefully selecting" or "We will" evaluate/present/highlight.

Check spelling. For example, Fig. 12 has error: functiion.

Author Response

  1. Abstract looks more like Introduction. It should be rewritten.

We thank the Reviewer for her/his suggestion. In the revised version of the paper, colored in red, the Abstract has been rewritten.

  1. There are too many self-references. Is it possible to reduce it? Why use self-citations in cases with more references [xx-xx]. I think that self-references before [26] are not necessary.

We thank the Reviewer for her/his suggestion. In the revised version of the paper some self-citations have been deleted.

  1. From the title, I expected more practical examples. You should modify the title to be better suited.

We thank the Reviewer for her/his suggestion. In the revised version of the paper the title has been changed. Now it is “Finite Differences for Recovering the Plate Profile in Electrostatic MEMS with Fringing Field”.

  1. Aims of the paper should be rewritten. For aims, it is not usual to have "By carefully selecting" or "We will" evaluate/present/highlight.

We thank the Reviewer for her/his suggestion. In the revised version of the paper, colored in red, corrections are visible.

  1. Check spelling. For example, Fig. 12 has error: functiion.

We thank the Reviewer for her/his suggestion. In the revised version of the paper, all typos have been removed.

Reviewer 2 Report (New Reviewer)

Please find the comments referring to the paper as an attachment.

Author Response

Please, see the attached .pdf file.

Round 2

Reviewer 2 Report (New Reviewer)

All comments of the reviewer have been included in the revised version of the paper. I recommend publication this paper in its present form.

This manuscript is a resubmission of an earlier submission. The following is a list of the peer review reports and author responses from that submission.

Round 1

Reviewer 1 Report

The authors described the behavior of an electrostatic elastic MEMS device with parallel plates by modeling with fringing field contribution by Pelesko-

Driscoll’s theory. They have some theoretical results and comparison with numerical results. However, they have no experimental results to prove the theoretical results are correct.

There are some comments for the authors.

1. Theoretical analysis is appreciated. But it is not suitable for publication without experimental results in an engineering journal. The authors can see the following papers.

1. C. H. Mastrangelo, and C. H. Hsu. Mechanical stability and adhesion of microstructures under capillary forces. I. Basic theory. Journal of Microelectromechanical Systems, 2(1), (1993), 33 –43.

2. C. H. Mastrangelo, and C. H. Hsu. Mechanical stability and adhesion of microstructures under capillary forces. II. Experiments. Journal of Microelectromechanical Systems, 2(1), (1993), 44 –55.

2.   Tension is very important in this research and is discussed in the results. However, there is no evidence to show its correctness. The authors must provide some experimental results to prove this.

3. In this work, Equations 64 is similar to Equations 65 and 66. However, the authors did not show their results are better or more significant than the results of references [48] and [49[. Therefore, it means the contribution of this paper is not significant.

Author Response

Reviewer #1

  1. Theoretical analysis is appreciated. But it is not suitable for publication without experimental results in an engineering journal. The authors can see the following papers.
  2. C. H. Mastrangelo, and C. H. Hsu. Mechanical stability and adhesion of microstructures under capillary forces. I. Basic theory. Journal of Microelectromechanical Systems, 2(1), (1993), 33 –43.
  3. C. H. Mastrangelo, and C. H. Hsu. Mechanical stability and adhesion of microstructures under capillary forces. II. Experiments. Journal of Microelectromechanical Systems, 2(1), (1993), 44 –55.
  4.  Tension is very important in this research and is discussed in the results. However, there is no evidence to show its correctness. The authors must provide some experimental results to prove this.
  5. In this work, Equations 64 is similar to Equations 65 and 66. However, the authors did not show their results are better or more significant than the results of references [48] and [49]. Therefore, it means the contribution of this paper is not significant.

We thank the Reviewer for her/his comments and suggestions. We emphasize that the paper was submitted to the Special Issue “Advances in Micro Electro Mechanical Systems: From MEMS to NEMS Devices” whose call for paper also encourages submission of theoretical papers where multi-physics models (such as electrostatic-elastic systems) are analyzed. We are aware that the work is purely theoretical. However, our results have been assessed with reference to experimental results available in literature. This remark puts the ground for an interesting scenario because the mathematical models that adhere to the physical reality of MEMS are extremely complex and do not allow analytical studies on them. Therefore, it is necessary to implement some simplifications in the geometry of the MEMS so that the model achieved can be studied mathematically: so, we did. It follows that the assessment obtained from the study of the model will give clear indications of the behavior of the MEMS device characterized by simplified geometry. Concerning the comparison of pull-in voltage formulations, in the revised version of the paper, numerous details have been added for comparison with experimental evidences know in the literature. Particularly:

*) some bibliographical references have been inserted regarding the analytical formulation of the effects due to the fringing field according to Pelesko and Driscoll’s theory;

*) some too long sentences have been reworded for a better understanding of the text;

*) the purposes of the work have been more clearly detailed in the introduction;

*) the figure that schematizes the studied device has been completed in all its parts;

*) the meaning of “industrially attractive limitation” has been specified in greater detail and clarity;

*) the choice of f(x) (which represents the dielectric profile of the deformable element) was physically motivated with greater incisiveness and clarity;

*) we have point out that the model studied represents a generalization of the model studied in previous works in which instead of the deformable plate there was a membrane;

*) we have shown that the model studied is dimensionless and all parameters have been defined in order of appearance;

*) we have underlined the fact that the functional links obtained between the mechanical tension of the deformable plate and the applied voltage are presented in dimensionalized form;

*) the adherences of the theoretical results discussed in this work with experimental evidence known in the literature have been detailed, with greater incisiveness and details.

Regarding the possible experimental confirmation of the results achieved for the mechanical tension, T, Figure 12 (included in the revised version of the paper) highlights

Reviewer #1

  1. Theoretical analysis is appreciated. But it is not suitable for publication without experimental results in an engineering journal. The authors can see the following papers.
  2. C. H. Mastrangelo, and C. H. Hsu. Mechanical stability and adhesion of microstructures under capillary forces. I. Basic theory. Journal of Microelectromechanical Systems, 2(1), (1993), 33 –43.
  3. C. H. Mastrangelo, and C. H. Hsu. Mechanical stability and adhesion of microstructures under capillary forces. II. Experiments. Journal of Microelectromechanical Systems, 2(1), (1993), 44 –55.
  4.  Tension is very important in this research and is discussed in the results. However, there is no evidence to show its correctness. The authors must provide some experimental results to prove this.
  5. In this work, Equations 64 is similar to Equations 65 and 66. However, the authors did not show their results are better or more significant than the results of references [48] and [49]. Therefore, it means the contribution of this paper is not significant.

We thank the Reviewer for her/his comments and suggestions. We emphasize that the paper was submitted to the Special Issue “Advances in Micro Electro Mechanical Systems: From MEMS to NEMS Devices” whose call for paper also encourages submission of theoretical papers where multi-physics models (such as electrostatic-elastic systems) are analyzed. We are aware that the work is purely theoretical. However, our results have been assessed with reference to experimental results available in literature. This remark puts the ground for an interesting scenario because the mathematical models that adhere to the physical reality of MEMS are extremely complex and do not allow analytical studies on them. Therefore, it is necessary to implement some simplifications in the geometry of the MEMS so that the model achieved can be studied mathematically: so, we did. It follows that the assessment obtained from the study of the model will give clear indications of the behavior of the MEMS device characterized by simplified geometry. Concerning the comparison of pull-in voltage formulations, in the revised version of the paper, numerous details have been added for comparison with experimental evidences know in the literature. Particularly:

*) some bibliographical references have been inserted regarding the analytical formulation of the effects due to the fringing field according to Pelesko and Driscoll’s theory;

*) some too long sentences have been reworded for a better understanding of the text;

*) the purposes of the work have been more clearly detailed in the introduction;

*) the figure that schematizes the studied device has been completed in all its parts;

*) the meaning of “industrially attractive limitation” has been specified in greater detail and clarity;

*) the choice of f(x) (which represents the dielectric profile of the deformable element) was physically motivated with greater incisiveness and clarity;

*) we have point out that the model studied represents a generalization of the model studied in previous works in which instead of the deformable plate there was a membrane;

*) we have shown that the model studied is dimensionless and all parameters have been defined in order of appearance;

*) we have underlined the fact that the functional links obtained between the mechanical tension of the deformable plate and the applied voltage are presented in dimensionalized form;

*) the adherences of the theoretical results discussed in this work with experimental evidence known in the literature have been detailed, with greater incisiveness and details.

Regarding the possible experimental confirmation of the results achieved for the mechanical tension, T, Figure 12 (included in the revised version of the paper) highlights the link obtained between the pull-in voltage and the distance, d, between the plates of the device; this link was found to be superimposable to experimental links known in literature. Then, by (72), also T, indirectly, is verified experimentally. In the revised version of the paper a Remark has been added to highlight this important aspect.

Finally, the suggested biography has been added in the reference list.

the link obtained between the pull-in voltage and the distance, d, between the plates of the device; this link was found to be superimposable to experimental links known in literature. Then, by (72), also T, indirectly, is verified experimentally. In the revised version of the paper a Remark has been added to highlight this important aspect.

Finally, the suggested biography has been added in the reference list.

Reviewer 2 Report

Dear Editor,

The authors have published similar research results in MDPI electronics “Deformable MEMS with Fringing Field: Models, Uniqueness Conditions and Membrane Profile Recovering”, the similar methods and topics. In this manuscript, the difference is just focusing on parallel plates. Therefore, compared with previous their work, I believe the originality is not enough. 

Author Response

Reviewer #2

The authors have published similar research results in MDPI electronics “Deformable MEMS with Fringing Field: Models, Uniqueness Conditions and Membrane Profile Recovering”, the similar methods and topics. In this manuscript, the difference is just focusing on parallel plates. Therefore, compared with previous their work, I believe the originality is not enough. 

We thank the Reviewer for her/his comments. We address to the Reviewer’s attention that the submitted paper is part of a line of research that the authors have been developing since many years and concerns the development of both membrane and deformable plate electrostatic MEMS models with reconstruction of the deformable element profile using numerical approaches. In particular, the submitted work concerns the extension of the results achieved for membrane MEMS to deformable plate MEMS. Regarding the paper “Deformable MEMS with Fringing Field: Models, Uniqueness Conditions and Membrane Profile Recovering”, it should be emphasized that the device in it considered is a membrane MEMS device in which the amplitude of the electric field has been shown to be proportional to the geometric curvature of the membrane itself. In contrast, in the submitted paper, the considered MEMS device exhibits parallel plates. The similarity of the results obtained between the submitted paper and the one already published derives essentially from the fact that it is necessary to find a link between the applied electrical voltage (which fixes the intended use of the device) and the mechanical tension of the deformable plate (which influences the choice of the material constituting the deformable plate). Furthermore, according the fringing field effects depend depended on the length/width ratio of the device, it was considered it useful to find an increase for the total electrostatic force (and, consequently, the electrostatic pressure) inside the device which depended not only on this ratio but also on the applied electrical voltage and on the mechanical tension of the deformable plate. Moreover, given that the model describing the behavior of the device does not allow for explicit solutions, we proceeded numerically to derive the profiles of the deformable plate in different operating conditions which, in the case of the submitted paper, were obtained through finite differences scheme. In contrast, in the paper already published, using Keller-Box schemes, shooting procedures and III/IV Stage Lobatto IIIa formulas were exploited. In any case, in the revised version of the paper, the following points have been implemented:

*) some bibliographical references have been inserted regarding the analytical formulation of the effects due to the fringing field according to Pelesko and Driscoll’s theory;

*) some too long sentences have been reworded for a better understanding of the text;

*) the purposes of the work have been more clearly detailed in the introduction;

*) the figure that schematizes the studied device has been completed in all its parts;

*) the meaning of “industrially attractive limitation” has been specified in greater detail and clarity;

*) the choice of f(x) (which represents the dielectric profile of the deformable element) was physically motivated with greater incisiveness and clarity;

*) we have point out that the model studied represents a generalization of the model studied in previous works in which instead of the deformable plate there was a membrane;

*) we have shown that the model studied is dimensionless and all parameters have been defined in order of appearance;

*) we have underlined the fact that the functional links obtained between the mechanical tension of the deformable plate and the applied voltage are presented in dimensionalized form;

*) the adherences of the theoretical results discussed in this work with experimental evidence known in the literature have been detailed, with greater incisiveness and details.

Reviewer 3 Report

The following general remarks can be highlighted:

1. The paper is not self-contained (independent). It relies excessively on previous works by the research team or by some other authors. It is therefore in my opinion an incremental paper. The added contribution is rather difficult to identify.

2. The paper is not reader-friendly. The engineering problem hypotheses are fuzzy as are the goals of the entire approach. It starts rather abruptly, without defining all the parameters and notations used involved in the model. For example, parameter T first appears on page 3, but its significance is revealed on page 4. Most of the parameters are indirectly characterized instead of being clearly defined (e.g., β, γ, δ, χ, λ). To be more specific, quote: “λ is a parameter depending on the applied voltage”. The following questions arise: How is it dependent? Its formula and computation procedure are not specified. Cannot the voltage itself be used, in the first place? Similar questions can be formulated. Another example: the significance of the tilde notation is not revealed. When using a notation, it has to be meaningful, i.e., to provide the reader with a certain amount of information. Is it the case for K1, K2, and F(x), all with the tilde notation placed above them? The same “fuzzy” characterization for the first two quantities is again used: they “are specific weight functions”. How are they defined and for what purpose? In which way are these parameters “specific”?

3. The paper totally lacks any unit of measurement for all the used quantities and functions. It is unusual for an engineering paper not to mention the units/subunits of measurement. As a result, the numerical values appearing in the formulas (e.g., (34) up to (39)) are totally irrelevant to the reader, because no dimensional representation can be attributed to these figures. A similar situation applies to the graph plotted in Figure 2, namely, there is no unit of measurement neither on the abscissa nor on the ordinate axes. Not specifying the unit of measurement for the involved parameters and quantities, makes it impossible for the equations to be checked dimensionally. For instance, “d* the minimum distance allowed between the two plates”, is a very small value being subtracted from 1 (dimensionless?) in the equality “u(x) = 1 d*”. If “1” is measured in meters then 1 >> d* and the function u(x) becomes irrelevant being practically constant, approximately 1 meter in value.

4. Last but not least, the paper has almost no reference concerning the “fringing field”. How is it computed: method, boundary conditions of the electrostatics problem, specific aspects, etc.? One would have expected an approach dedicated to “fringing field” computation aspects.

In more detail, the following remarks can be made:

a. The phase “Today, theoretical modeling […]” of Section 1 is too long and confusing. Its meaning is difficult to understand.

b. The depiction of the device shown in Figure 1 has no parameter attached to the vertical double-headed arrow. It is probably the distance separating the two parallel plates.

c. What do you mean by “industrially attractive limitation”?

d. Is the f(x) relationship given in (18) has some clearer physical significance (a quantitative one) than “the center of the MEMS device is notoriously the most unstable, because the influence of the electrostatic field is very strong, while the influence of the supporting is weak”? The introduction in (18) of the power law looks arbitrary in the absence of an electromechanical quantitative justification.

Author Response

We ask the Reviewer to consider the attached .docx file.

Round 2

Reviewer 1 Report

Although theoretical analysis is encouraged to submit to the journal, the theoretical results need to be verified by experimental results. It does not found in the paper. Therefore, it is not known whether the theoretical analysis matches the physical conditions practically or not. And it provides poor significance. 

Reviewer 3 Report

The paper has been significantly improved. The authors answered most of my concerns. I have no further remarks.